# Stratospheric Downward Wave Reflection Events Modulate North American Weather Regimes and Cold Spells

Gabriele Messori[1, 2], Marlene Kretschmer[3], Simon H. Lee[4], and Vivien Wendt[5,*]

[1]Department of Earth Sciences and Centre of Natural Hazards and Disaster Science (CNDS), Uppsala University, Uppsala, Sweden.
[2]Department of Meteorology and Bolin Centre for Climate Research, Stockholm University, Stockholm, Sweden.
[3]Department of Meteorology, University of Reading, Reading, UK.
[4]Department of Applied Physics and Applied Mathematics, Columbia University, New York, NY, USA.
[5]Institute for Solar-Terrestrial Physics, German Aerospace Center (DLR), Neustrelitz, Germany.
[*]née Matthias

**Correspondence:** Gabriele Messori (gabriele.messori@geo.uu.se)

**Abstract.** The Arctic stratospheric polar vortex is an important driver of mid-latitude winter cold spells. One proposed coupling mechanism between the stratospheric polar vortex and the troposphere is upward-propagating planetary waves being reflected downward by the polar vortex. However, while the wave reflection mechanism is well-documented, its role in favouring cold spells is still under-explored. Here, we analyse such stratospheric wave reflection and its impact on the tropospheric circulation and surface temperatures over North America in winter. We present a physically interpretable regional stratospheric wave reflection detection metric, and identify the tropospheric circulation anomalies associated with prolonged periods of wave reflection, which we term *reflection events*. In particular, we characterise the tropospheric anomalies through the lens of North American weather regimes. Stratospheric reflection events show a systematic evolution from a Pacific Trough regime — associated on average with positive temperature anomalies and a near-complete absence of anomalously cold temperatures in North America — to an Alaskan Ridge regime, which favours low temperatures over much of the continent. The most striking feature of the stratospheric reflection events is thus a rapid, continental-scale decrease in temperatures. These emerge as continental-scale colds spells by the end of the reflection events. Stratospheric reflection events are thus highly relevant in a tropospheric predictability perspective.

## 1 Introduction

Notwithstanding rapidly rising global temperatures, wintertime cold spells continue to have a large impact on society. The North American continent has experienced an ostensibly large number of cold spells during recent winters, including repeated episodes during the winters of 2013/2014[1] (Trenary et al., 2015; Van Oldenborgh et al., 2015), 2017/2018[2] (Matthias and Kretschmer, 2020), 2018/2019[3] (Lee and Butler, 2020; Lillo et al., 2021) and 2020/2021[4] (Doss-Gollin et al., 2021). While

---

[1]"Un Québec Froid dans un Monde Chaud". La Presse. Retrieved July 14, 2022.
[2]"Dangerously Cold Temperatures Grip Midwest as 2018 Begins". Time.com. Retrieved January 4, 2022.
[3]"Polar vortex death toll rises to 21 as US cold snap continues". BBC News. Retrieved January 4, 2022.
[4]"These US cities had the coldest morning in decades – with some reaching all-time record lows". CNN. Retrieved January 4, 2022.

cold spells are expected to decrease in frequency globally (Screen, 2014; Van Oldenborgh et al., 2019), some studies have argued that this decrease may not be as rapid as would be expected from the increase in global mean temperatures (Gao et al., 2015; Cohen et al., 2021).

The drivers of wintertime North American cold spells are multifarious. There is a broad literature focusing on the mid-latitude tropospheric dynamics, including modes of climate variability (e.g. Assel, 1992; Linkin and Nigam, 2008; Loikith and Broccoli, 2014; Budikova et al., 2021), planetary wave patterns (e.g. Harnik et al., 2016; Rudeva and Simmonds, 2021) and regional-to-continental scale weather regimes (e.g. Robertson and Ghil, 1999; Vigaud et al., 2018; Lee et al., 2019) or large-scale meteorological patterns (e.g. Grotjahn et al., 2016; Messori et al., 2016; Faranda et al., 2020). Other studies have highlighted the role of remote forcing in driving some of the mid-latitude tropospheric patterns that in turn favour surface cold spells. Next to tropical signals (e.g. Ropelewski and Halpert, 1986; Hartmann, 2015; Watson et al., 2016; Scaife et al., 2017; Dai and Tan, 2019), the Arctic stratospheric polar vortex has been reported in this context. The latter denotes a fast-flowing westerly airstream forming in boreal winter in the northern high-latitude stratosphere. Variability in the vortex strength projects onto variability in the mid-latitude tropospheric circulation (e.g. Castanheira and Barriopedro, 2010; Davini et al., 2014; Hitchcock and Simpson, 2014) and has been related to extreme winter weather in different geographical regions, including North America (Baldwin and Dunkerton, 2001; Kolstad et al., 2010; Kretschmer et al., 2018a; Monnin et al., 2021; King et al., 2019; Domeisen and Butler, 2020).

Two different coupling mechanisms between the stratospheric polar vortex and the tropospheric circulation have been proposed, which may be interpreted as different facets of planetary wave-zonal flow interaction. The first focuses on the interaction between upward-propagating or internally-generated planetary waves and the stratospheric zonal flow, whereby wave activity convergence in the stratosphere decelerates the westerly flow of the stratospheric polar vortex. This induces a negative stratospheric Northern Annular Mode (NAM), whose signal can then propagate down to the troposphere (Matsuno, 1971; Baldwin and Dunkerton, 2001; Dunn-Sigouin and Shaw, 2015; Kidston et al., 2015). In extreme cases, the stratospheric westerlies can reverse direction (known as 'major sudden stratospheric warmings'; SSWs), often followed by a prolonged negative tropospheric NAM (which projects onto the negative phase of the North Atlantic Oscillation) and increased surface cold spells in the mid-latitudes, particularly northern Eurasia (Garfinkel et al., 2017; Kretschmer et al., 2018a, b; Zhang et al., 2020). The second stratosphere-troposphere mechanism involves upward-propagating tropospheric waves being reflected downward by the polar vortex, which thereby exerts an indirect influence on the tropospheric circulation (Harnik and Lindzen, 2001; Perlwitz and Harnik, 2003, 2004; Shaw et al., 2010; Shaw and Perlwitz, 2013). In contrast to SSWs, this mechanism has been associated with a strong stratospheric polar vortex and a positive phase of the North Atlantic Oscillation (Shaw et al., 2014; Dunn-Sigouin and Shaw, 2015; Kidston et al., 2015; Lubis et al., 2016; Rupp et al., 2022). However, we note that the impacts of SSWs may also be interpreted through the lens of downward propagation of wave anomalies (Zhang et al., 2020). Recently, stratospheric wave reflection has been linked with significant cold-air outbreaks over North America (Kodera et al., 2016; Kretschmer et al.,

2018a; Matthias and Kretschmer, 2020; Guan et al., 2020; Millin et al., 2022).

Despite the increasing body of evidence supporting the role of downward wave reflection in favoring North American cold-spells, this mechanism has garnered less attention than weak polar vortex events and major SSWs. Potential reasons include the lack of a full understanding of the dynamical processes underlying wave reflection (e.g. Harnik (2009); Lubis et al. (2017, 2018)) and the difficulty in diagnosing reflection events (see the discussion in Matthias and Kretschmer, 2020). Building upon results from wave geometry diagnostics (Perlwitz and Harnik, 2003, 2004), cluster analysis (Kretschmer et al., 2018a),
and early work on reflection events by Kodera et al. (2008), Matthias and Kretschmer (2020) introduced a simple index to identify wave reflection events based on anomalous lower-stratospheric poleward eddy-heat flux over Siberia and Canada. Consecutive days with a high regional reflection index were shown to be followed by North Pacific blocking events, favouring cold spells over North America. While Matthias and Kretschmer (2020) discussed in detail the dynamical properties of individual cold spells during the winter of 2018/19, a systematic documentation of the role of stratospheric wave reflection in
leading to tropospheric circulation anomalies and cold spells in North America is missing from the literature.

    Here, we combine the stratospheric and tropospheric perspectives to investigate how stratospheric wave reflection is associated with tropospheric weather regimes and surface cold spells. Motivated by the widespread scientific coverage and public interest elicited by recent high-impact cold spells over North America (see references above), we focus on this geographical
region. We extend upon previous work (e.g. Kretschmer et al., 2018a; Matthias and Kretschmer, 2020) by providing an updated regional reflection event definition that is both relatively straightforward to compute and physically interpretable in terms of the dynamical properties of wave reflection. Moreover, we provide a systematic analysis of the tropospheric circulation associated with reflection-driven North American cold spells. We thus seek to trace the whole mechanistic chain from stratospheric wave reflection, to tropospheric weather regimes, to the resulting surface temperature anomalies.

**2   Data and Methods**

We base our analysis on data from the European Centre for Medium-Range Weather Forecasts (ECMWF) ERA5 reanalysis (Hersbach et al., 2020). We use daily data covering the period from 1 December 1979 to 31 March 2021, and focus on an extended winter season covering the months of December, January, February and March (DJFM). All climatologies are defined as the average of a 15-day centered mean of the same calendar days for all years in the dataset. For example, the climatological
temperature of 12 December is the average temperature during 5–19 December of all years from 1979 to 2020. Anomalies are then computed as daily deviations from this climatology. For 2-m temperature we additionally smooth the anomalies with a 9-day running mean, which gives greater prominence to persistent temperature anomalies. The 2-m temperature anomalies are further linearly detrended using area-mean 2-metre land temperature over North America (30–72.5 °N, 190–305 °E, the same domain as shown in Fig. 4).


Wave reflection events are identified based on meridional eddy-heat fluxes in the lower stratosphere (see Sect. 3), and only events starting and ending within DJFM are analysed. Therefore, events which occur outside this season are not considered here.

To identify cold spells, we use 2-m temperature (also referred to as *surface temperature* in the text) at 0.5° horizontal resolution over the domain 40–55 °N, 260–290 °E. This corresponds to a region experiencing anomalously low temperatures for cold spells affecting the eastern portion of the continent (such as the 2017/2018 cold spells) as well as those extending to the more southerly regions (such as during the 2020/2021 winter), while avoiding too large a domain that may lead to cancellation of anomalies and aliasing. A cold spell day is defined as a local minimum in area-averaged 2-m temperature anomalies or a local maximum in the number of gridpoints within the domain below the 5th percentile of the local temperature anomaly distribution. The local minima/maxima are defined by centering on an 11-day period. This is equivalent to imposing a minimum 5-day separation between consecutive cold spells, and again seeks to minimise aliasing.

The North American weather regimes are computed using 1.5° horizontal resolution data, as they are intended to represent continental-scale patterns (Fig. A1). Moreover, the calculation procedure (see Sect. 5) is performed in a truncated empirical orthogonal function (EOF) space, making the results largely insensitive to reasonable variations in horizontal resolution.

To test the robustness of composite-means to the sampling, statistical significance is assessed by bootstrapping the set of chosen events 10,000 times (with replacement) to generate 95% confidence intervals on the mean. For geographical maps and pressure–longitude maps, we control for multiple testing by applying the Benjamini and Hochberg false detection rate (FDR) procedure, with $\alpha_{FDR} = 0.1$ to yield an approximate global significance level of $\alpha = 0.05$ (Wilks, 2016).

## 3 Definition of stratospheric reflection events

We aim to identify stratospheric wave reflection events over the North Pacific which exhibit an influence on tropospheric circulation and surface weather over North America. For this purpose, we build upon and update the reflection index from Matthias and Kretschmer (2020). As discussed here and in the following sections, this simple index can detect regional wave reflection.

The reflection index, $RI$, is defined as the difference in anomalous poleward eddy-heat fluxes in the lower stratosphere over Siberia (Sib) and Canada (Can):

$$RI = (v'T')^*_{Sib} - (v'T')^*_{Can}. \tag{1}$$

Here $v$ and $T$ denote meridional wind velocity and temperature at 100 hPa, computed on a 1° horizontal grid, and the primes denote deviations from the zonal-mean. Regional area-weighted averages are calculated over Siberia (Sib, 140°–200°E, 45°–

75°N) and Canada (Can, 230°–280°E, 45°–75°N). The asterisks indicate that the regional time-series have been standardized by removing the daily mean and subsequently dividing by the daily standard deviation. Note that the regional boxes used for the index have been modified compared to Matthias and Kretschmer (2020), to represent the regions of strongest positive (Siberia) and negative (Canada) anomalous values of $v'T'$ (Fig. 1a, b).

In Matthias and Kretschmer (2020), reflection events were then defined as days where $RI$ exceeds 1.5 for at least 10 consecutive days. Here, motivated by the subsequent analyses (see Figs. 1, 2) we use the lower threshold of 1 to diagnose reflection *days*, but keep the persistence criterion of 10 days to select reflection *events* (Fig. 3). Higher RI thresholds of 1.5 and 2 and persistence thresholds between 7 and 14 days lead to qualitatively similar surface temperature anomaly patterns (not shown). In the following, we will indeed show that this simple index and the applied event criterion (i.e., $RI > 1$ for at least 10 consecutive days), have a physical basis and capture downward wave reflection events affecting North American winter weather.

For this index to be a dynamically interpretable representation of downward wave reflection over the North Pacific, the following three conditions need to be fulfilled (Perlwitz and Harnik, 2003; Matthias and Kretschmer, 2020):

1. There is upward wave propagation over Siberia, i.e. $(v'T')_{Sib} > 0$.

2. There is downward wave propagation over Canada, i.e. $(v'T')_{Can} < 0$.

3. There is a reflective surface in the stratosphere.

To show that our index fulfills the first two conditions, we compute histograms of $(v'T')_{Sib}$ and $(v'T')_{Can}$ (Figure 1 c, d). According to linear theory, positive absolute values of zonal-mean $v'T'$ indicate upward wave propagation, while negative values indicate downward propagation – assuming the wave-activity density is positive definite. This relationship holds for zonal-mean values, but it was shown in Matthias and Kretschmer (2020) (additionally using the vertical component of the Plumb fluxes) that this can also be derived for the regional averages used here. During almost all winter days, $(v'T')_{Sib}$ is positive (Fig. 1c), while $(v'T')_{Can}$ takes both positive and negative values (Fig. 1c). During days where $RI > 1$, wave propagation over the Canadian domain-average is instead almost exclusively negative (i.e., $(v'T')_{Can} < 0$), Fig. 1d), except for 28 days in our sample ($\sim 1.5\%$ of all reflective days). During these days, however, values are locally strongly negative. We therefore deem this discrepancy negligible for the purpose of our analysis. Moreover, upward wave propagation over Siberia is particularly pronounced during the selected days (Fig. 1d). Collectively, this shows that days when $RI > 1$ represent an enhancement of the climatological state: there is *increased upward* wave propagation over Siberia and *enhanced downward* wave propagation over Canada.

To next assess under which conditions $RI$ fulfills the third criterion (i.e., the presence of a reflective surface), we follow Perlwitz and Harnik (2003). Using the quasigeostrophic equation of conservation of potential vorticity, they showed that negative vertical wind shear in the stratosphere corresponds to the formation of a vertical reflective surface. Figure 2 shows the

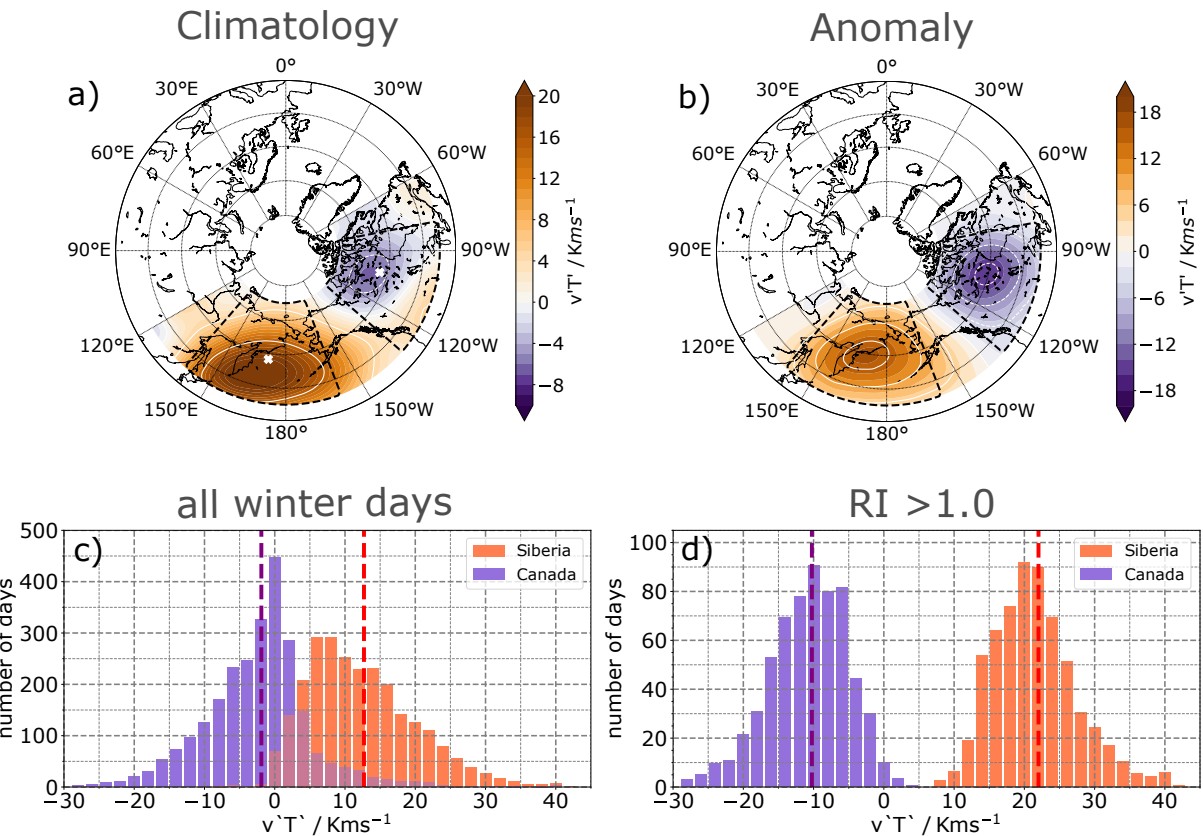

**Figure 1.** a) Climatology of the wintertime local daily meridional heat flux $v'T'$ at 100 hPa in the Siberian and Canadian sectors. The white crosses mark the average centers of mass of the meridional heat flux in the Siberian and Canadian sectors during reflection events. b) Composite anomaly of the local daily meridional heat flux at 100 hPa during reflection events. Histograms of daily meridional heat flux $v'T'$ at 100 hPa averaged for the Canadian (blue) and Siberian (red) sectors for (c) all winter days and (d) only for days when $RI > 1.0$. The vertical dashed lines represent the averages over all days. The data covers the DJFM seasons from 1979/80 to 2020/21

stratospheric vertical zonal-mean zonal wind profile, averaged over $60 - 80°$N. When $RI > 1$, there is negative vertical shear of the zonal wind in the stratosphere above $\sim$10 hPa, meaning that the zonal-mean zonal winds weaken with height (green line in Figure 2a). This indicates the formation of a reflective surface (Perlwitz and Harnik, 2003). In contrast, the zonal-mean zonal wind velocities increase with height during days when $RI < 1$ (blue line), similar to the climatological winds (red line).

Figure 2b further shows the vertical wind profile for different $RI$ thresholds ranging from 0 to 3, with the thresholds larger than $RI > 0$ showing negative vertical shear of the zonal wind. In fact, we find that the higher the threshold, the stronger the curvature of the wind profile (see also Fig. 8 in Matthias and Kretschmer (2020) and discussion therein). Meridional profiles of the zonal wind at 30 hPa averaged over the Canadian sector (Fig. A2) further show an increased curvature in mid and high latitudes for $RI > 1$ days relative to both climatology and $RI < 1$ days, supportive of the existence of a meridional waveguide

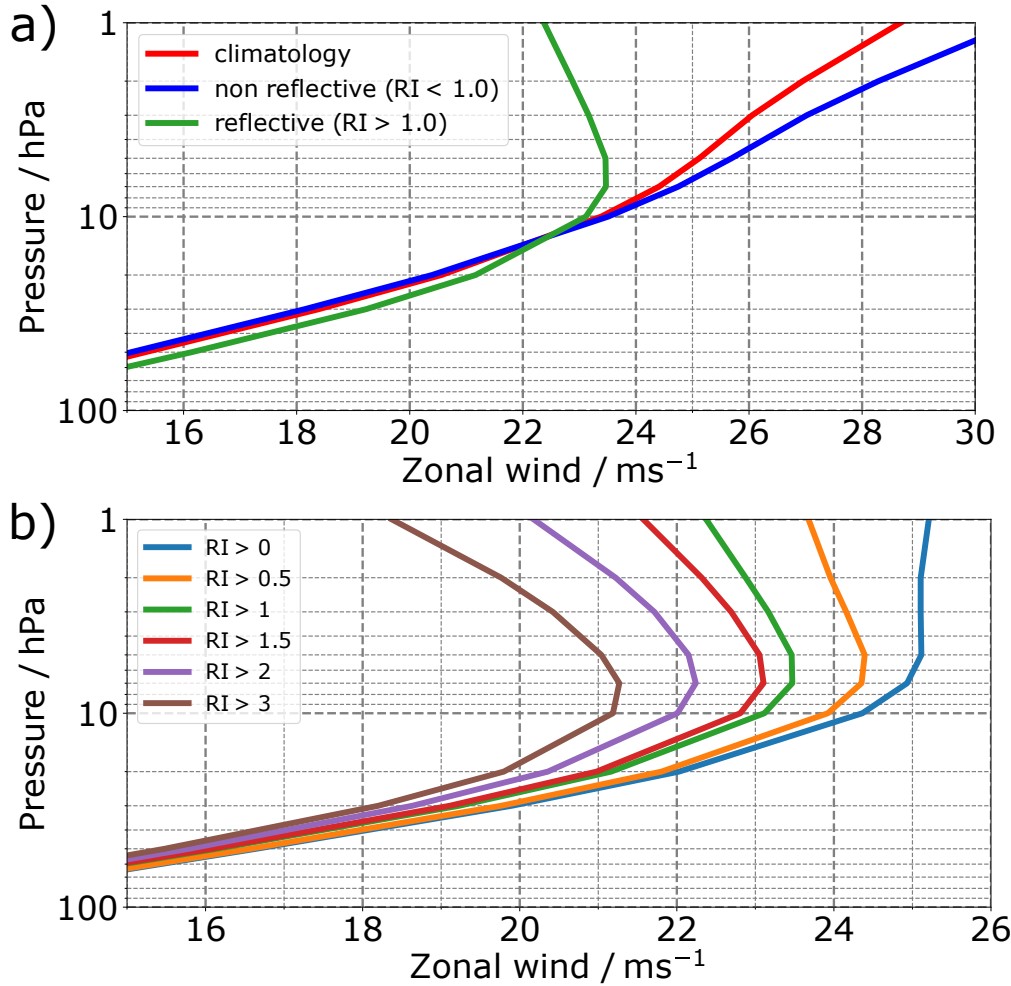

**Figure 2.** Vertical zonal-mean zonal wind profiles (averaged over $60-80°$N) in the stratosphere. a) Climatology of all winter days (red line), days when $RI > 1$ (green line) and days when $RI < 1$ (blue line). b) Same as (a), but for days when $RI$ exceeds different thresholds.

during reflection days.

     Finally, following Matthias and Kretschmer (2020) we apply a persistence criterion of 10 days to sub-select reflection events from the set of all reflective days. This results in a total of 44 events over the studied time-period (Table A1). The onset and end dates of a given event refer to the first and last day when $RI > 1$, respectively. The peak date is the day when the largest

index value is obtained. On average, there is just over 1 event per winter. Most events occur during January and February, during the climatological peak of polar vortex variability (Fig. A3c). The number of reflective days per winter varies between

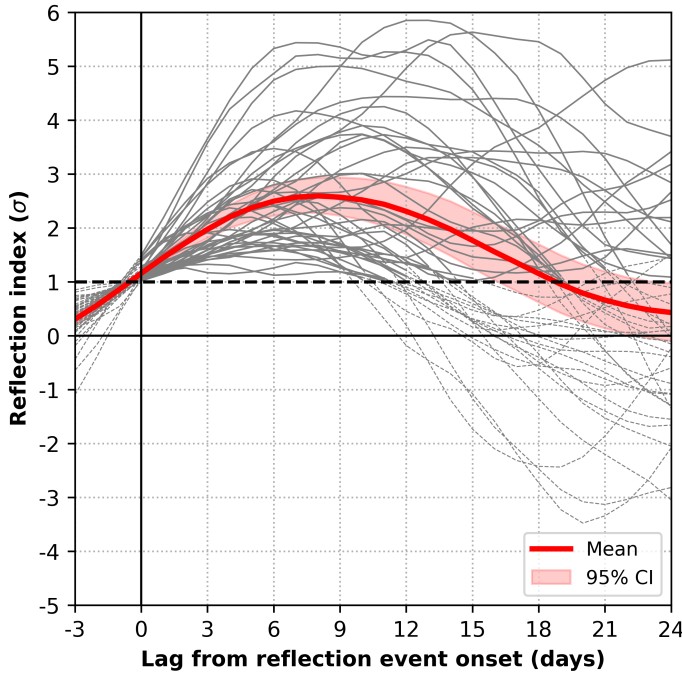

**Figure 3.** Evolution of the reflection index (grey lines) for the 44 identified reflection events (i.e., $RI > 1$ for at least 10 consecutive days) as a function of days from event onset. Lines are dashed where the threshold is not met. The thick red line denotes the average over all events and the black horizontal dashed line indicates the threshold of $RI = 1$. Shading indicates a 95% confidence interval on the mean assessed as described in Sect. 2.

none and almost 70, and does not always show a direct correspondence to the number of reflection events (Fig. A3a, b). Figure 3 shows the evolution of the reflection index for the 44 reflection events (grey lines), as a function of days from event onset, as well as the average over all events (thick red line). There is a large spread in the magnitude and persistence of reflection events, with some events lasting more than 4 weeks and reaching RI values of close to 6. The median event duration is 20 days, with a maximum of 66 days (event starting 2 January 2016). Whilst the minimum duration is set at 10 days, the average $RI$ in our 44-event sample is significantly greater than 1 for over two weeks. We henceforth focus on the 24 days following the reflection event onset, as this both captures the full duration of the typical events and is the maximum lag for which data fall within DJFM for all 44 events. The onset of these reflection events is associated with an anomalously strong stratospheric polar vortex (negative Z10 anomalies over the central Arctic, Fig. A4), consistent with the expected favorable conditions for wave reflection. Only a modest vortex stretching (as described in Cohen et al. (2021)) and a return of the vortex strength to climatological levels is observed at positive lags, as the reflection events develop (Fig. A5).

In summary, all three conditions characterising downward stratospheric wave reflection are fulfilled for $RI > 1$. We additionally find evidence for the existence of a waveguide during these same days. Our simple index is thus suitable to identify

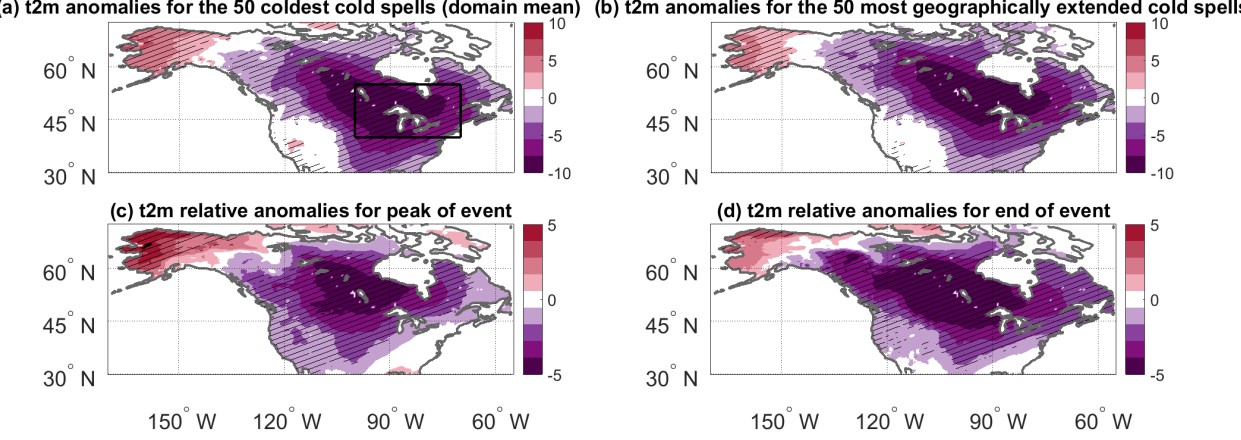

**Figure 4.** Composite-mean 2-metre temperature (t2m) anomalies (K) for: (a) the 50 cold spells with the lowest area-averaged t2m anomaly over 40–55 °N, 260–290 °E (black box); and (b) the 50 cold spells with the most grid-points below the local 5th percentile of t2m anomalies over the same domain. In both cases, a minimum separation of 5 days is enforced between different cold spells. Composite t2m anomalies (K) relative to onset of reflection events for (c) the peak; and (d) the end of all reflection events. Hatching denotes statistically significant anomalies, assessed as described in Sect. 2. Note that the colourbar range differs between panels (a, b) and (c, d).

days of wave reflection over the North Pacific. In the following, we analyse the tropospheric evolution associated with the 44 reflection events.

## 4   North American Cold Spells and Stratospheric Reflection Events

The 50 most extreme North American cold spells (see definition in Sect. 2) have a coherent geographical footprint, corresponding to an elongated region of anomalously low temperatures stretching from central-western Canada to the south-eastern seaboard of the continent. Alaska and the west coast display near-zero or positive anomalies (Fig. 4a, b). This is consistent with the patterns observed in earlier studies (e.g. Van Oldenborgh et al., 2015; Messori et al., 2016). The picture is very similar regardless of whether one defines cold spell severity based on area-averaged temperature anomalies or number of grid-points

below the 5th percentile of the local temperature anomaly distribution (cf. Fig. 4a, b). An analysis quantifying the frequency of local negative or extremely negative (<5th percentile) temperature anomalies shows a similar pattern (Fig. A6).

We next consider the t2m anomalies associated with the 44 stratospheric reflection events as defined in Sect. 3. On average, at event onset there are strong positive t2m anomalies across North America (Fig. 5a). This changes by the peak of the reflection

events (i.e., the time the $RI$ reaches its maximum), when weak negative t2m anomalies begin to emerge in the central-northern part of the continent (Fig. 5b). Finally, by the end of the reflection events negative anomalies dominate across the central-

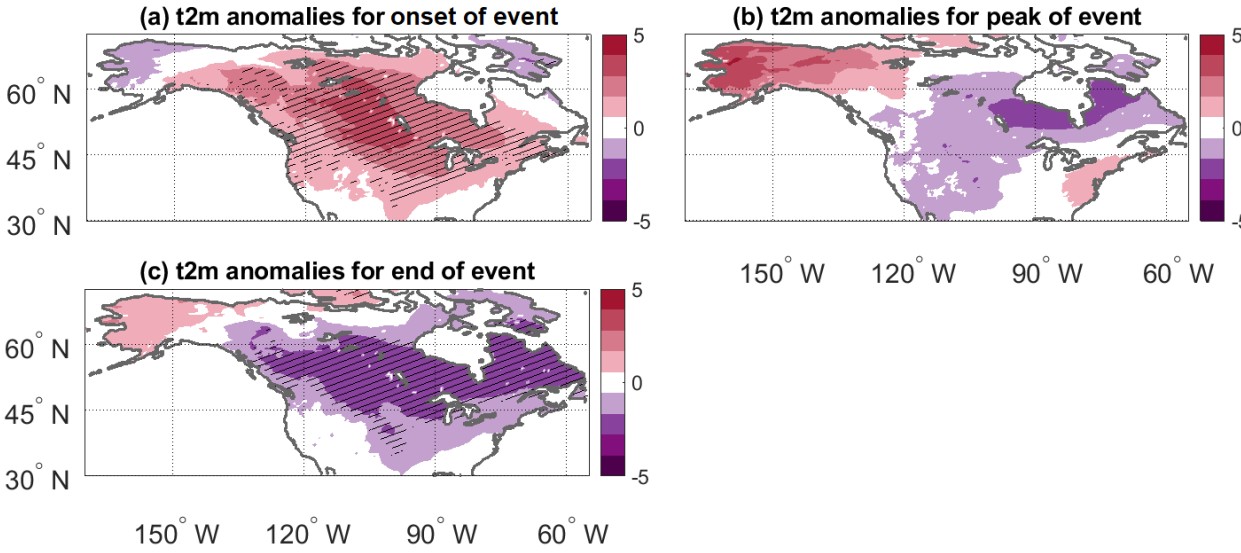

**Figure 5.** Composite-mean 2-metre temperature (t2m) anomalies at the (a) onset, (b) peak and (c) end of the reflection events. Hatching denotes statistically significant anomalies, assessed as described in Sect. 2.

eastern parts of the continent, indicating the typical geographical footprint of North American cold spells, albeit with smaller magnitude than the average of the 50 coldest events in the dataset (cf. Fig. 4a,b, with Fig. 5c). A similar picture emerges by considering calendar day lags relative to onset date of the reflection events (Fig. 6). The reflection events are characterised by a

gradual shift from positive temperature anomalies at the onset of the event to negative temperature anomalies towards the end of the event.

The stratospheric reflection events thus correspond to a drop in temperatures across most of North America. Indeed, taking as reference the t2m at the onset of the reflection events, negative anomalies in the range of -4 to -5K dominate across a

large part of North America already by the peak of the reflection events. By the end of the events, the anomalies strengthen further, although they remain weaker in magnitude than the 50 coldest spells (cf. Fig. 4a,b with panels c,d in the same figure). Only Alaska and the south-eastern corner of the domain show neutral or weakly positive anomalies (Fig. 4d). A similar picture is obtained if one considers the composite t2m anomaly difference between day 10 of the events and their onset (Fig. 6f).

Individual stratospheric reflection events may also be separated according to the associated temperature anomalies over the target domain 40–55 °N, 260–290 °E relative to their onset date (Table A1). By 10 days after the event onset, almost two-thirds of the events display area-averaged t2m anomalies < -0.5 K. By the end date, over two-thirds of the events display t2m anomalies < -0.5 K. Figure A7 shows the timeline of mean temperature anomalies relative to event onset as a function of lag

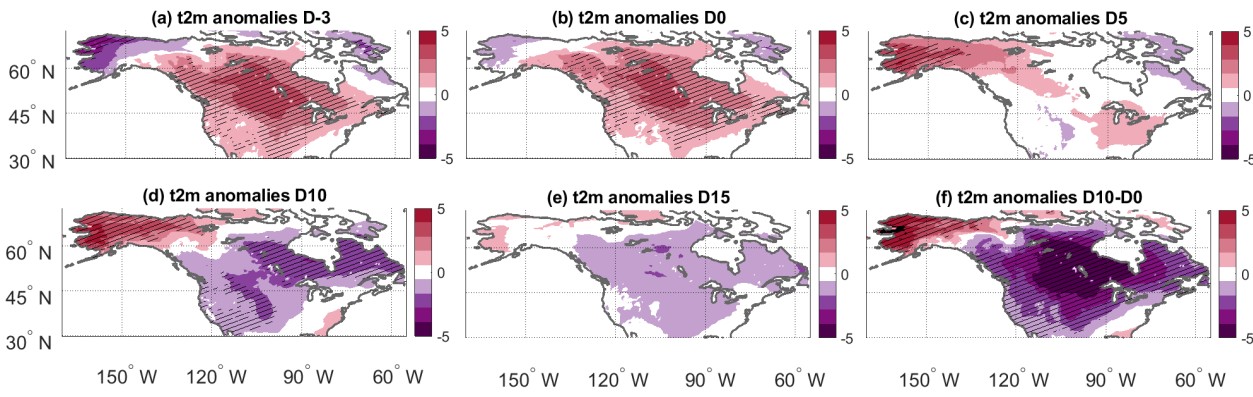

**Figure 6.** Composite-mean 2-metre temperature (t2m) anomalies at various lags relative to the reflection event onset. (f) Average difference between the t2m anomalies on day 10 and day 0 (i.e., (d)-(b)). Hatching denotes statistically significant anomalies, assessed as described in Sect. 2.

from onset for different classes of stratospheric reflection events (see Fig. caption).


## 5 Tropospheric Dynamics Linking Reflection Events to North American Cold Spells

The above analysis suggests a connection between stratospheric reflection events and a large-scale lowering of surface temperatures across the North American continent. To better understand the underlying dynamical mechanisms, we analyse the tropospheric large-scale patterns associated with the stratospheric reflection events.


The composite-mean 500 hPa geopotential anomalies (Z500) for the 44 identified reflection events are shown in Fig. 7 (a–e) for lags between -3 and +15 days relative to the onset of the events. At lags of -3 and 0 days, there is a significant anomalous trough in the northeastern Pacific and across Alaska. A ridge anomaly is present across most of the contiguous USA and southern Canada, centred near the Hudson Bay. This pattern resembles the Pacific Trough weather regime (Vigaud et al., 2018; Lee

et al., 2019; Robertson et al., 2020) and a positive North Pacific Oscillation (NPO) (Linkin and Nigam, 2008). By lag +5 days, the anomalous trough has been replaced by an anomalous ridge – a westward progression of the anomaly present over central North America previously. By day +10, the pattern from lag 0 has reversed: there is now an anomalous ridge over Alaska and an anomalous trough over the Hudson Bay extending down to the southwestern USA, resembling the Alaskan Ridge regime (Vigaud et al., 2018; Lee et al., 2019) with some negative NPO characteristics. This pattern persists, albeit slightly weaker, up

to day +15.

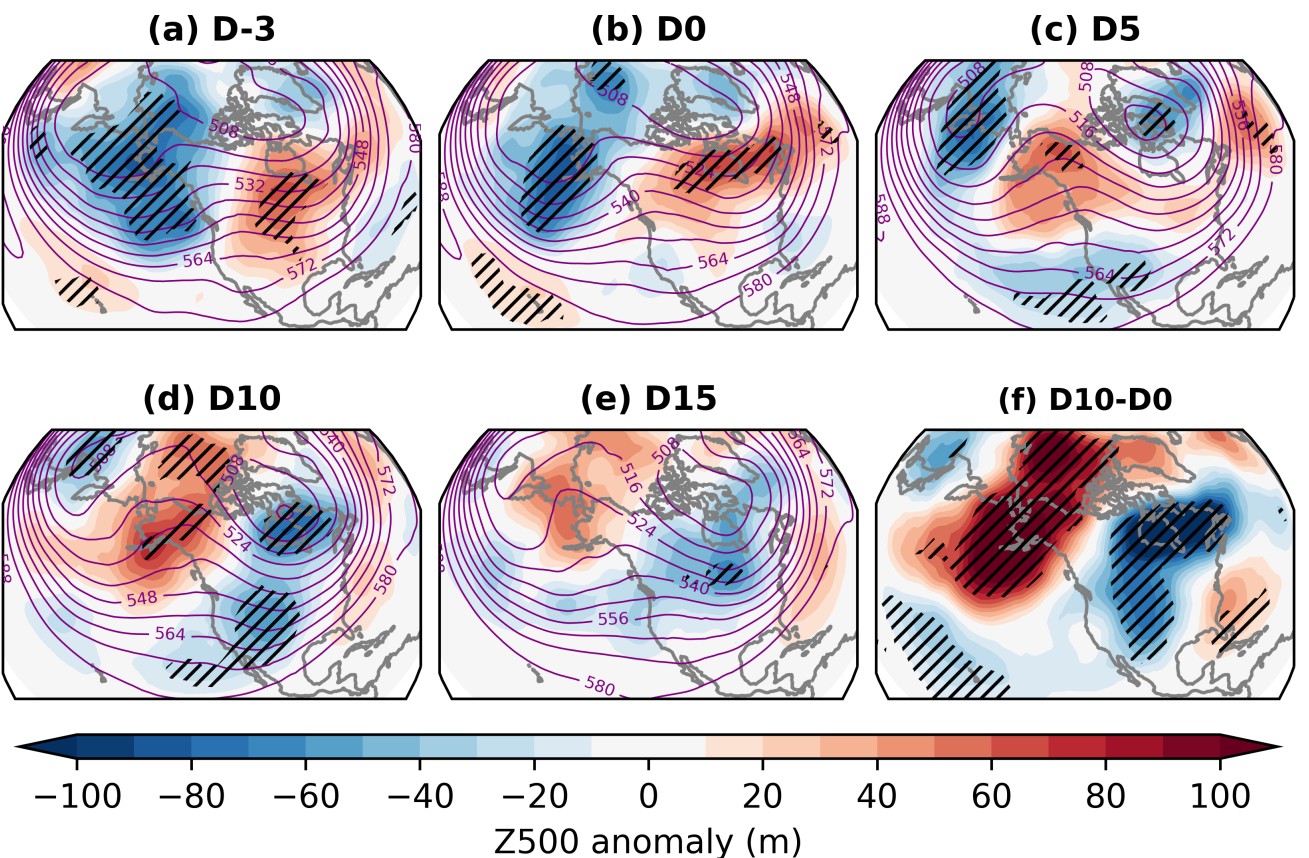

**Figure 7.** (a–e) Composite-mean 500 hPa geopotential height (Z500) (purple contours, dam) and anomalies (shading, m) at various lags relative to the reflection event onset. (f) Average difference between the Z500 anomalies on day 10 and day 0 (i.e., (d)-(b)). Hatching denotes statistically significant anomalies, assessed as described in Sect. 2.

There is, thus, a marked inversion of the large-scale Z500 pattern on a timescale of $\sim$10 days. To illustrate the Z500 *tendency* between day 0 and day +10, Figure 7 (f) shows the mean difference in Z500 anomalies between these lags. The resultant pattern is similar to the mean anomalies at day +10, but of greater amplitude and with more widespread statistical significance. This suggests that the tendency in the flow pattern is of greater magnitude and more robust than the resultant anomaly. The ridge and trough nodes of the tendency pattern are respectively located in Bristol Bay in the eastern Bering Sea (57°N, 198°E) and the Hudson Bay (58.5°N, 274.5°E), and are very close to the centres of the respective ridge and trough anomalies that characterise the Alaskan Ridge regime (c.f. Fig. A1c).

Due to the similarity between the Z500 patterns associated with the stratospheric reflection events and previously-defined North American weather regimes, we propose an interpretation of the evolution of the tropospheric circulation from a regimes

perspective. We adopt four North American weather regimes (Fig. A1) following Lee et al. (2019), namely (in order of climatological frequency): Arctic High (ArH), Arctic Low (ArL), Alaskan Ridge (AkR) and Pacific Trough (PT). The choice of four regimes is considered optimal for this domain (Vigaud et al., 2018). These are determined using $k$-means clustering in the space spanned by the leading 12 principal components (PCs) of the daily Z500 anomalies in the region 180–330 °E, 20–80 °N during DJFM. The choice of 12 PCs is made to emphasise the larger, slowly-varying states (see discussion in Robertson et al. (2020)), and explains around 80% of the variance, though the regimes are primarily determined by only the leading 3 EOFs (Lee et al., 2022). Each day is then assigned to a regime by determining the nearest cluster centroid (by Euclidean distance) in 12-dimensional PC space. The regimes are by definition persistent, such that the most likely transition between two consecutive days is from a regime to itself.

Figure 8 (a–d) shows the lagged evolution of the proportion of days assigned to each regime across all 44 reflection events. There are large, significant and opposing changes in the frequency of the AkR and PT regimes. The AkR regime is unlikely immediately prior to the onset of the reflection event. It then approximately triples in frequency within the first five days after the event onset to become slightly more frequent than climatology and peaks in frequency around days 9 to 12. At all subsequent positive lags shown here, the regime is more frequent than in the days before the event onset (Fig. 8c). Meanwhile, the PT regime is initially over twice as likely as climatology before and at the onset of the reflection event, with a rapid decline in frequency over the following 10 days. By day 12, the PT regime is almost 50% less likely than climatology (or, alternatively, more than three times less likely than at the event onset, Fig. 8d). The ArH and ArL regimes show weaker changes in frequency as the reflection events progress, and mostly display near-climatological occurrence. The overall picture of a transition from PT to AkR as the reflection events develop is supported by the regime transition statistics (Fig. 9). The transition from PT to AkR is climatologically the least likely (4.4%) of the 12 possible transitions (Fig. 9a) suggesting it is not simply a typical tropospheric evolution following a PT regime. Furthermore, PT to AkR is one of the two transitions showing the closest correspondence with wave reflection (Fig. 9b; the other being ArH to AkR, whilst AkR self-transition is also significantly associated with reflection).

The regime evolution can also be viewed in terms of the average normalised regime projection, which enables an analysis of continuous shifts in the flow pattern which do not necessarily alter the discrete regime attribution. This can be considered in a similar way to the PC timeseries of an EOF, but without the associated orthogonality or variance partitioning constraints. The projection is defined using a method based on the weather regime index of Michel and Rivière (2011). First, the Z500 field for each day is truncated to the leading 12 EOFs and then projected onto the composite-mean for all days assigned to each regime. The resulting timeseries are then normalised by their means and standard deviations. Figure 8 (e–h) shows the average lagged evolution of this quantity for each regime across the 44 reflection events. As with the regime frequency, there is little average change to the projection onto the ArH and ArL regimes (although the latter shows a small but insignificant increase), but large changes to the projections onto the AkR and PT regimes. On average, there is an increase in the projection onto the AkR regime by $\sim 1\ \sigma$, and a corresponding $\sim 1\ \sigma$ decrease in the projection onto the PT regime. The evolution of the AkR and PT projections almost mirror each other, with both switching from around +/-0.5 $\sigma$ to -/+0.5 $\sigma$ in $\sim 1$ week after the onset of

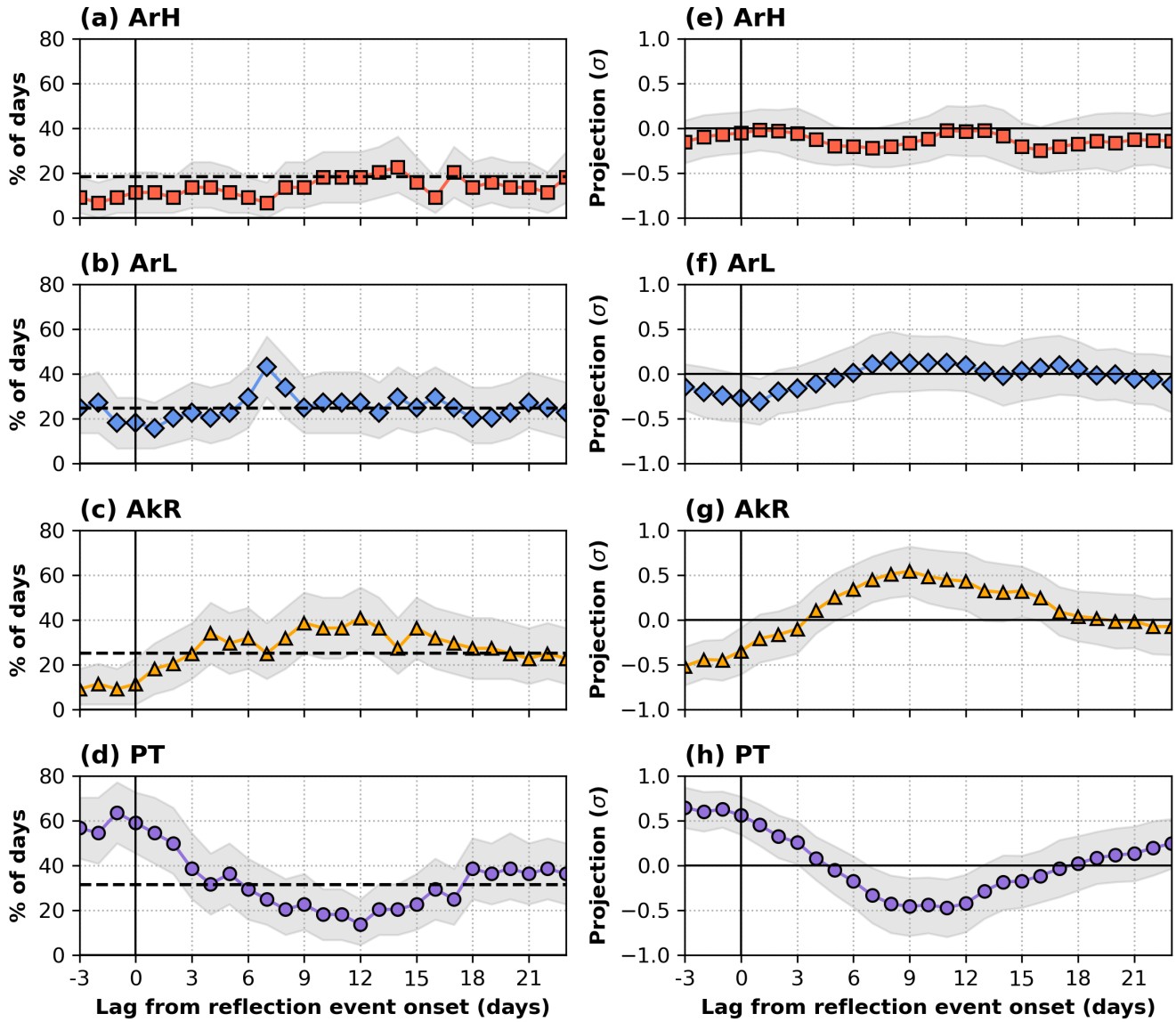

**Figure 8.** (a–d) Proportion of days in the 44-event sample assigned to each regime. The horizontal dashed lines indicate the climatological DJFM frequency of each regime. (e–h) Mean normalised projection onto each regime for the 44 events. Grey shading indicates 95% confidence intervals assessed as described in Sect. 2.

the reflection event, becoming significantly different from zero from around day 7 through to day 12.

Overall, the regime-based evolution is in good agreement with the evolution of the full Z500 field shown in Figure 7. These
results confirm that the chief tropospheric impact of the stratospheric reflection events is of favouring a strong pattern *tendency*

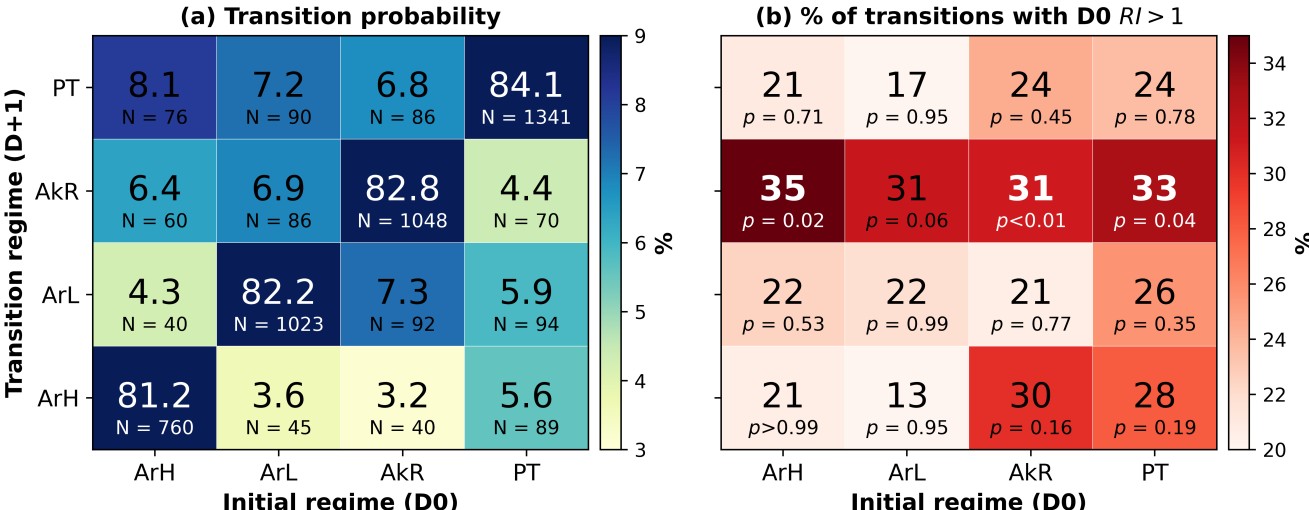

**Figure 9.** (a) Transition matrix for the four North American weather regimes. For each initial regime (x-axis), the numbers in each column denote the observed probability (expressed as a percentage; columns sum to 100) of persisting in the same regime (white font) or transitioning into a different regime. The total number of instances of each transition (N) is also shown. (b) Percentage of each transition pathway which occurs with RI>1 on the day prior to the transition (D0), expressed as percentages. P-values indicate the estimated probability of obtaining a statistic greater than the observed value by chance, assessed by randomly re-sampling all DJFM days 10,000 times (without replacement) using the observed sample sizes for each transition pathway. Statistics significant at the one-sided 5% significance level are in bold white font.

– i.e., away from PT and toward AkR – rather than simply leading to the onset of an AkR regime. Furthermore, whilst the occurrence of the ArH regime is strongly modulated by the lower-stratospheric zonal mean winds (Lee et al., 2019, 2022), our results suggest it is only weakly modulated by the occurrence of reflection events. In a similar but opposite sense, whilst the occurrence of the AkR regime is largely insensitive to the strength of the lower-stratospheric zonal mean winds (Lee et al., 2019, 2022), it shows (alongside PT) the largest sensitivity to stratospheric wave reflection. The contrasting relationship between these different forms of stratospheric variability and the ArH and AkR regimes is similar to the differing tropospheric response to 'absorbing' and 'reflecting' SSWs described in Kodera et al. (2016). When taken together, these results demonstrate the importance of multiple aspects of stratospheric variability for modulating North American weather and climate.

The different weather regimes are associated with distinct surface anomalies. When considering extremely cold t2m anomalies (defined as before as anomalies below the 5th percentile of the local anomaly distribution), the AkR regime clearly dominates across central-eastern North America, with a footprint that closely resembles that of the 50 coldest cold spells (cf. Fig. A6c, d, with Fig. 10c). A similar picture emerges if one considers the fraction of days within each regime associated with negative t2m anomalies (cf. Fig. A6a, b, with Fig. A8c). On the contrary, PT corresponds to virtually no extremely cold t2m anomalies and to very few negative t2m anomalies in the central and eastern parts of North America (Figs. 10d, A8d). A

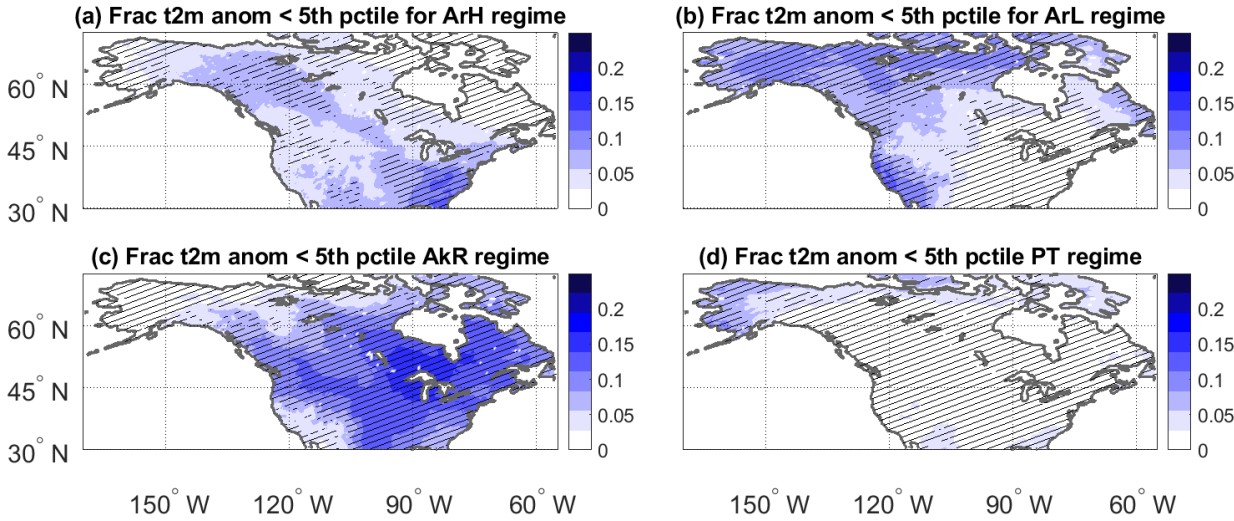

**Figure 10.** Fraction of t2m anomalies (K) below the local 5th percentile for days in the: (a) ArH, (b) ArL, (c) AkR and (d) PT weather regime. Hatching denotes statistically significant anomalies, assessed as described in Sect. 2.

timeline of the lagged occurrence of the different weather regimes relative the peak of the 50 coldest spells in North America, confirms the important role of AkR in driving strong negative t2m anomalies, and of PT in suppressing the occurrence of strong negative t2m anomalies (Fig. A9). This is in agreement with Lee et al. (2019), who found that AkR and PT are respectively associated with the warmest and coldest average anomalies over most of North America.


The reflection events therefore begin in a PT-like configuration, with widespread positive t2m anomalies in North America. As they evolve, the troposphere transitions to an AkR-type configuration, associated with a rapid drop in temperatures and moderate to large negative t2m anomalies over central-eastern North America.

## 6 Discussion and Conclusions

We have identified a set of stratospheric events affecting the North American continent, by developing an event definition that is both straightforward to compute and physically interpretable in terms of reflection of upward-propagating Rossby waves. These events correspond to relatively strong negative temperature anomalies across a large part of North America, but their most striking feature is that they are systematically associated with a sharp lowering of the surface temperatures. Indeed, the onset of the stratospheric reflection events sees widespread positive temperature anomalies, which rapidly drop to negative
temperature anomalies on synoptic timescales. A similar role of stratospheric wave reflection in leading to a drop in temperatures in central-eastern North America was recently discussed by Guan et al. (2020), in the context of an intraseasonal North American temperature variability mode. Cold spells can have severe socio-economic impacts (e.g. Doss-Gollin et al., 2021),

and rapid temperature swings are a hazard in their own right. They can for example lead to widespread ecosystem impacts in the case of *false spring* events (e.g. Kral-O'Brien et al., 2019), or more generally to unexpectedly large damages even in the
absence of extreme absolute anomalies (Casson et al., 2019). The association between stratospheric reflection events and rapid continental-scale surface temperature drops is therefore highly relevant from an impacts-based perspective.

The onset of the stratospheric reflection events is associated with a strong stratospheric polar vortex, which is then partially stretched by the development of the Alaskan/Aleutian ridge typical of the AkR regime. The vortex stretching is associated with
a geographical shift of the negative 10 hPa geopotential height anomalies and a return of the vortex strength to climatological levels. Indeed, at positive lags there is almost no signal in the 10 hPa 60 °N zonal-mean zonal wind anomaly (Fig. A5b), which suggests that the reflection events do not induce a specific 10 hPa zonal-mean zonal wind response as they develop. Nonetheless, significant negative 10 hPa geopotential height anomalies persist over the polar region for over 10 days following the onset of the reflection events (Fig. A4). These resemble the wave-1 asymmetric mode of winter stratospheric variability
identified by Ding et al. (2022), whose footprint is indeed associated with a strengthening of the Alaskan/Aleutian ridge and a surface cooling signal over North America. Our results further echo the evolution of the intraseasonal temperature variability mode discussed in Guan et al. (2020), which displays a strengthening of the Alaskan ridge and upstream trough associated with stratospheric wave reflection.

We also analysed whether the reflection events overlap with the occurrence of SSWs (Table A1), finding that SSWs occur during only 9 of the 44 reflection events. An additional 3 reflection events are preceded by an SSW within 20 days of their onset. Of the 9 events coinciding with an SSW, 5 do not show a substantial drop in surface temperature anomalies over the cold spell domain we analyse here (Fig. 4a), i.e. they are "warm" or "neutral" events. Moreover, in only 3 of these 9 events does an SSW occur within the first 15 days, which are the lags on which our analysis focuses (and of these, none occur within the first
9 days). Hence, there is minimal correspondence between our reflection events and SSWs. This is dynamically relevant, as any reflection events occurring during SSWs may involve over-reflection rather than linear reflection (Harnik and Heifetz, 2007), and their evolution and surface impacts may thus be viewed as a mixture between a reflective SSW and linear reflection events (e.g. Kodera et al., 2016).

In agreement with the strengthened stratospheric polar vortex observed during the reflection events, we find that reflection events preferentially occur during the westerly phase of the Quasi-Biennial Oscillation (QBOw). Specifically, 30 of the 44 reflection events onset during a QBOw month, and 31 peak during a QBOw month. This is in contrast to the preferential occurrence of SSWs and a weakened stratospheric polar vortex during the easterly QBO, albeit with strong low-frequency modulation (Holton and Tan, 1980; Lu et al., 2008). The relationship between the QBO and reflection events implies potential
forewarning of the propensity for the stratosphere to favour reflection events at lead-times beyond a season, which – alongside the ability of models to capture this apparent teleconnection – deserves further study.

We interpret the surface impacts of the stratospheric reflection events through the lens of North American weather regimes. We use a four-regime classification, where each day can be assigned to a regime based on its large-scale 500 hPa geopotential height anomalies, and a normalised strength of the projection onto the regime can be computed. The stratospheric reflection events show a systematic evolution from a Pacific Trough regime – associated on average with positive temperature anomalies and a near-complete absence of anomalously cold temperatures in North America – to an Alaskan Ridge regime, which favours low temperatures over much of the continent. A case-by-case depiction of the weather regime evolution during the individual stratospheric reflection events identified in our analysis is shown in Figure 11. The bulk of the events show what might be considered a *canonical* evolution: a progression from a Pacific Trough to an Alaskan Ridge regime. However, the timing and duration of the regime transitions exhibits large case-by-case variability, and it is possible to identify some unusual cases. For example, the stratospheric reflection events occurring on 4 January 1995 and 14 December 2020 progress in the opposite sense, from an Alaskan Ridge to a Pacific Trough. These events were associated with increasing surface temperatures across North America, and are classified as *warm* events (Table A1). In other words, a specific stratospheric evolution does not deterministically dictate a specific tropospheric response, which may be due to inter-event differences in both the troposphere and stratosphere. The same applies to the tropospheric response to major SSWs (e.g. Beerli and Grams, 2019; Domeisen et al., 2020; Davis et al., 2022).

Nonetheless, the robust statistical link between wave reflection events and surface temperature drops across North America make these events potentially relevant in a predictability perspective. Indeed, since the seminal work of Baldwin and Dunkerton (2001), the stratosphere has often been singled out as a powerful source of information for medium- to extended-range tropospheric forecasts, notably in the case of SSWs (e.g. Karpechko, 2018; Sigmond et al., 2013). In this perspective, a study of the predictability of stratospheric wave reflection events would provide a valuable proof-of-concept for their use as predictors of tropospheric impacts. The comparatively frequent occurrence of these events (on average once per extended winter season) versus major SSWs (on average about two out of every three winters) supports their potential usefulness in a predictability context. This perspective could fruitfully be combined with known tropospheric predictors of North American cold spells (e.g. Harnik et al., 2016) to verify whether these are two aspects of the same large-scale circulation anomalies or to some degree independent drivers of surface temperature anomalies.

On longer timescales, it is important to understand whether the occurrence of reflection events may be modulated by natural and anthropogenic forcings. In the literature, there is evidence for the role of both in leading to large-scale circulations that favour downward wave reflection events and the associated surface impacts (Lubis et al., 2016; Omrani et al., 2016; Lu et al., 2017; Lubis et al., 2018). Although not investigated in this study, such modulation of reflection events could be a significant control on future cold spell occurrence.

A second open question pertains the physical mechanisms connecting stratospheric wave reflection to tropospheric anomalies, and in particular whether the downward planetary wave reflection transfers enough energy to elicit a direct tropospheric

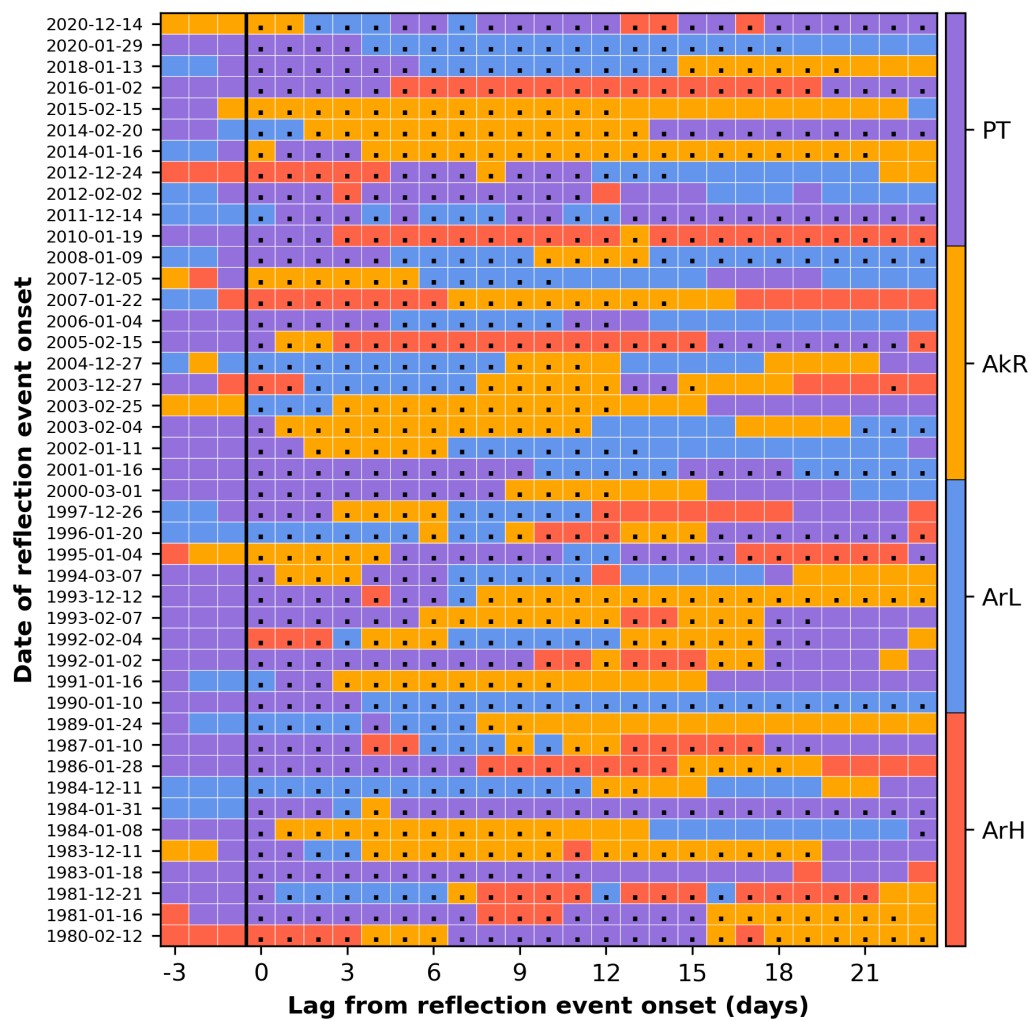

**Figure 11.** Evolution of the daily regime assignment during the 44 reflection events. Days with $RI > 1$ are shown with dots.

and surface response. As alternative mechanism one may hypothesise an indirect impact of stratospheric wave reflection on the troposphere via a modulation of baroclinicity and hence forcing of synoptic transient eddies (e.g. Smy and Scott, 2009; Thompson and Birner, 2012; Lubis et al., 2016, 2018).

A third avenue requiring further investigation is whether the causal chain leading to the tropospheric impacts starts from the wave reflection events, or whether there are tropospheric precursors to the stratospheric reflection events that may allow even longer-range statistical predictions. Several studies have demonstrated the existence of tropospheric precursors to variability in the strength of the stratospheric polar vortex (e.g. Bao et al., 2017; Kretschmer et al., 2018b; White et al., 2019; Peings, 2019) primarily through modulating tropospheric sources of upward-propagating wave activity. Zhang et al. (2020) have also

shown the importance of sea-ice conditions as amplifiers of SSW impacts on North American cold spells. The large-scale surface and tropospheric circulation anomalies preceding the onset of the reflection events we analyse here – characterised by the presence of a PT regime and a warm North America – suggest that a similar argument may hold for the latter events. Indeed, the eddy height field and Plumb wave activity flux associated with the reflection events (Fig. 12) shows how the pattern associated with the PT regime is linked to a westward-tilting and upward-propagating wave packet which is detected by the Siberian node of our reflection event definition. It thus appears likely that a pattern similar to PT is required to initiate the enhanced upward propagation from Siberia. At the same time, PT is the most frequent regime (31.6% of DJFM days), and thus far more common than reflection events (which account for just under 20% of DJFM days). This is reminiscent of the relationship between blocking and SSWs, blocking being by far the more common of the two (e.g. Peings, 2019). This implies a key role of the stratosphere in guiding the waves downward, further evidenced by the structure of the wave field showing an eastward-tilting wave structure over the Canada node and the vertical component of the Plumb wave activity flux (Fig. 12). How exactly this propagation impacts the tropospheric circulation, including the location and sinuosity of the eddy-driven jet stream, also requires further investigation.

While the full causal and mechanistic chain leading from the onset of a stratospheric reflection event to surface temperature anomalies still remains to be unraveled, our analysis shows that the reflection events are robustly associated to widespread and severe wintertime surface temperature decreases across North America. Therefore, we suggest that forecasts of wave reflection are likely to be a useful tool for extended-range prediction of North American weather and for understanding tropospheric forecast uncertainty.

*Data availability.* ERA5 data is freely available from the Copernicus Climate Change Service via https://cds.climate.copernicus.eu/. The North American weather regime data and the reflective index timeseries will be made available through GitHub upon acceptance of the manuscript.

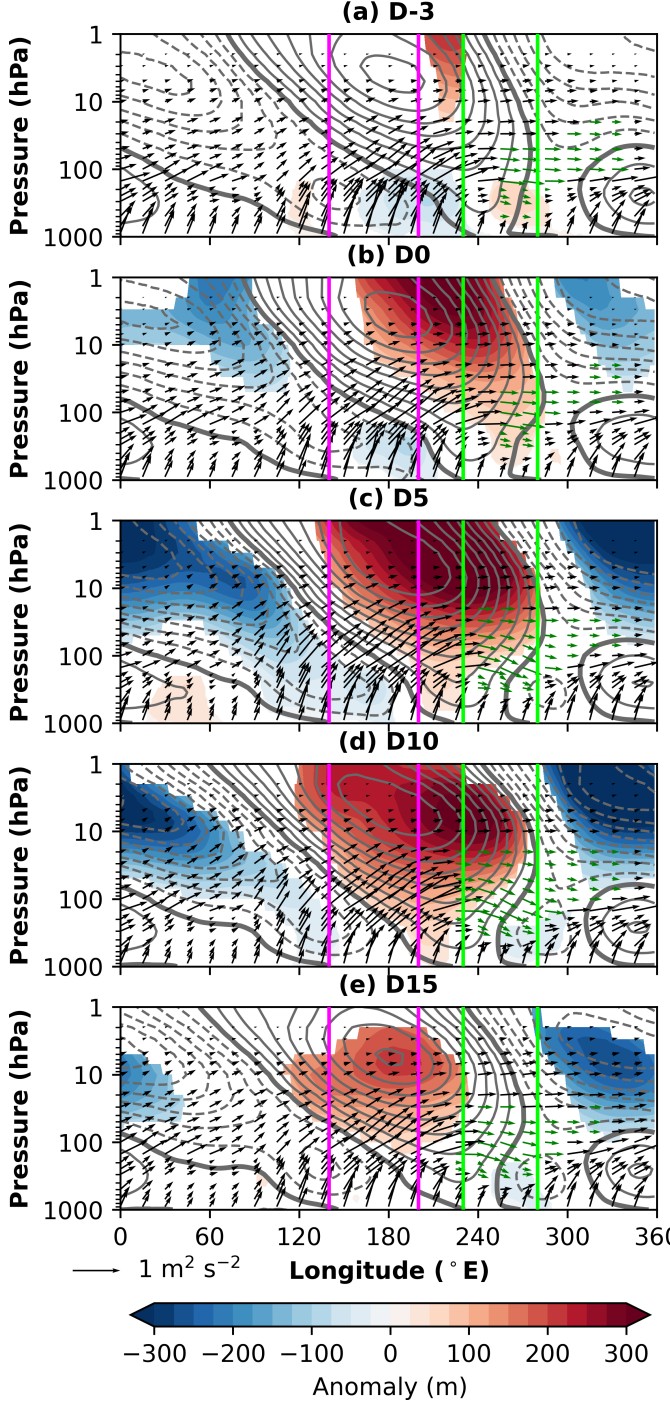

**Figure 12.** Evolution of the mean vertical wave structure associated with reflection events. Grey contours denote the average 40–80° N eddy geopotential height field (contours every 100 m, dashed negative, zero contour thickened). Shading denotes significant departures of the eddy height field from climatology, with significance assessed as described in Sect. 2. Pink vertical lines delineate the longitudinal range of the Siberian box (140–200° E), and green vertical lines delineate the longitudinal range of the Canadian box (230–280° E). Arrows denote the vertical and zonal components of the 40–80° N average Plumb wave activity flux, calculated over wavenumbers 1-3 (Plumb, 1985). Green arrows indicate the vertical component is downward. Arrows are scaled following Millin et al. (2022).

## Appendix A:  Additional Tables and Figures

Table A1: Onset and end dates, temperature classification and SSW coincidence for the 44 selected stratospheric reflection events. An SSW is considered to coincide with a reflection event if the dates of the two events overlap. We used the SSW catalogue from NOAA Chemical Sciences Laboratory (2020) and the additional event described in Lee (2021). For the SSWs in NOAA Chemical Sciences Laboratory (2020) only those present in at least two reanalysis products were counted.

| Event No. | Onset Date | End Date | Class | SSW |
|:---:|:---:|:---:|:---:|:---:|
| 1 | 12-Feb-1980 | 09-Mar-1980 | Neutral | Y |
| 2 | 16-Jan-1981 | 07-Feb-1981 | Warm | N |
| 3 | 21-Dec-1981 | 11-Jan-1982 | Cold | N |
| 4 | 18-Jan-1983 | 29-Jan-1983 | Warm | N |
| 5 | 11-Dec-1983 | 30-Dec-1983 | Cold | N |
| 6 | 08-Jan-1984 | 18-Jan-1984 | Cold | N |
| 7 | 31-Jan-1984 | 02-Mar-1984 | Warm | Y |
| 8 | 11-Dec-1984 | 24-Dec-1984 | Cold | N |
| 9 | 28-Jan-1986 | 15-Feb-1986 | Warm | N |
| 10 | 10-Jan-1987 | 29-Jan-1987 | Cold | Y |
| 11 | 24-Jan-1989 | 02-Feb-1989 | Cold | N |
| 12 | 10-Jan-1990 | 10-Feb-1990 | Warm | N |
| 13 | 16-Jan-1991 | 26-Jan-1991 | Cold | N |
| 14 | 02-Jan-1992 | 20-Jan-1992 | Cold | N |
| 15 | 04-Feb-1992 | 23-Feb-1992 | Cold | N |
| 16 | 07-Feb-1993 | 26-Feb-1993 | Cold | N |
| 17 | 12-Dec-1993 | 27-Jan-1994 | Cold | N |
| 18 | 07-Mar-1994 | 18-Mar-1994 | Cold | N |
| 19 | 04-Jan-1995 | 14-Feb-1995 | Warm | N |
| 20 | 20-Jan-1996 | 08-Mar-1996 | Cold | N |
| 21 | 26-Dec-1997 | 07-Jan-1998 | Cold | N |
| 22 | 01-Mar-2000 | 13-Mar-2000 | Cold | N |
| 23 | 16-Jan-2001 | 18-Feb-2001 | Neutral | Y |
| 24 | 11-Jan-2002 | 24-Jan-2002 | Cold | N |
| 25 | 04-Feb-2003 | 15-Feb-2003 | Cold | N |
| 26 | 25-Feb-2003 | 09-Mar-2003 | Cold | N |

| 27 | 27-Dec-2003 | 11-Jan-2004 | Cold | Y |
|----|-------------|-------------|---------|---|
| 28 | 27-Dec-2004 | 07-Jan-2005 | Warm | N |
| 29 | 15-Feb-2005 | 18-Mar-2005 | Cold | N |
| 30 | 04-Jan-2006 | 16-Jan-2006 | Warm | N |
| 31 | 22-Jan-2007 | 05-Feb-2007 | Cold | N |
| 32 | 05-Dec-2007 | 15-Dec-2007 | Warm | N |
| 33 | 09-Jan-2008 | 01-Mar-2008 | Cold | Y |
| 34 | 19-Jan-2010 | 14-Feb-2010 | Cold | Y |
| 35 | 14-Dec-2011 | 17-Jan-2012 | Warm | N |
| 36 | 02-Feb-2012 | 13-Feb-2012 | Cold | N |
| 37 | 24-Dec-2012 | 07-Jan-2013 | Neutral | Y |
| 38 | 16-Jan-2014 | 06-Feb-2014 | Cold | N |
| 39 | 20-Feb-2014 | 17-Mar-2014 | Cold | N |
| 40 | 15-Feb-2015 | 27-Feb-2015 | Neutral | N |
| 41 | 02-Jan-2016 | 08-Mar-2016 | Cold | N |
| 42 | 13-Jan-2018 | 02-Feb-2018 | Warm | N |
| 43 | 29-Jan-2020 | 16-Feb-2020 | Cold | N |
| 44 | 14-Dec-2020 | 29-Jan-2021 | Warm | Y |

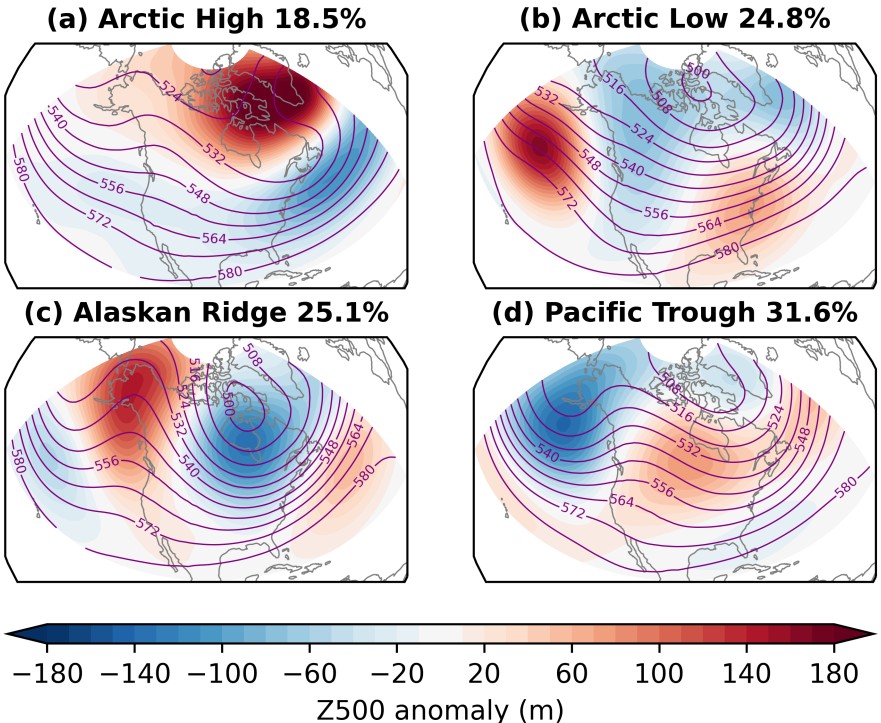

**Figure A1.** Composite-mean 500 hPa geopotential height (Z500) (purple contours, dam) and anomalies (shading, m) for all days assigned to each of the four North American wintertime weather regimes during DJFM 1979–2021. The proportion of days assigned to each regime are shown as percentages.

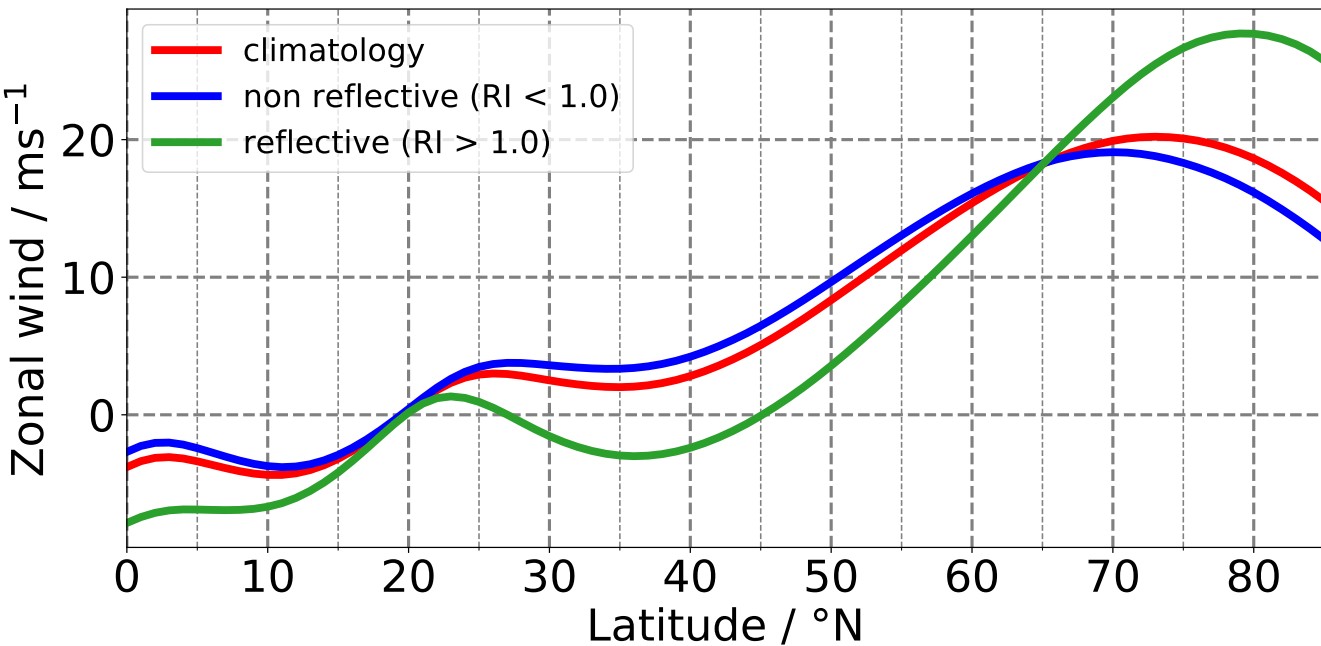

**Figure A2.** Meridional profiles of the zonal wind averaged over 230–280 °E (Canadian sector) at 30hPa. Cimatology of all winter days (red line), days when RI > 1 (green line) and days when RI < 1 (blue line).

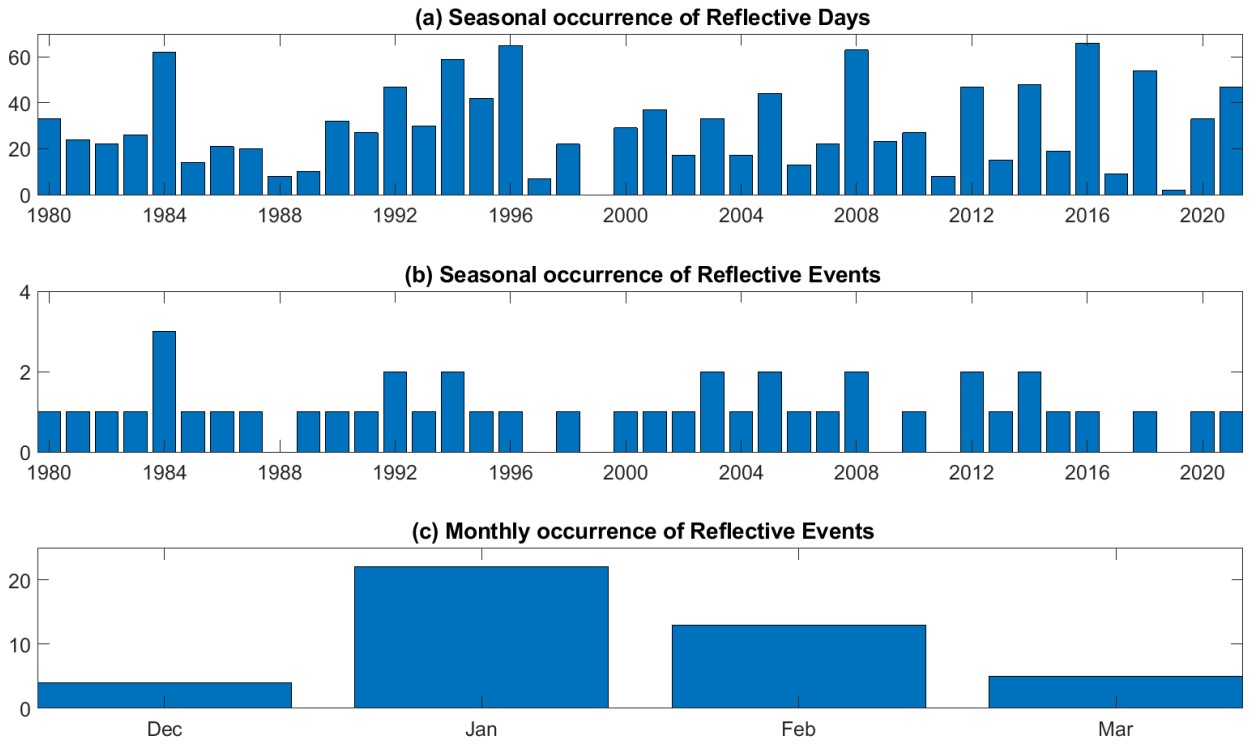

**Figure A3.** Seasonal occurrence of reflective days ($RI > 1.0$) and seasonal (b) and monthly (c) occurrence of stratospheric reflection events from December 1979 to March 2021. For reflective events, the date of maximum $RI$ is used.

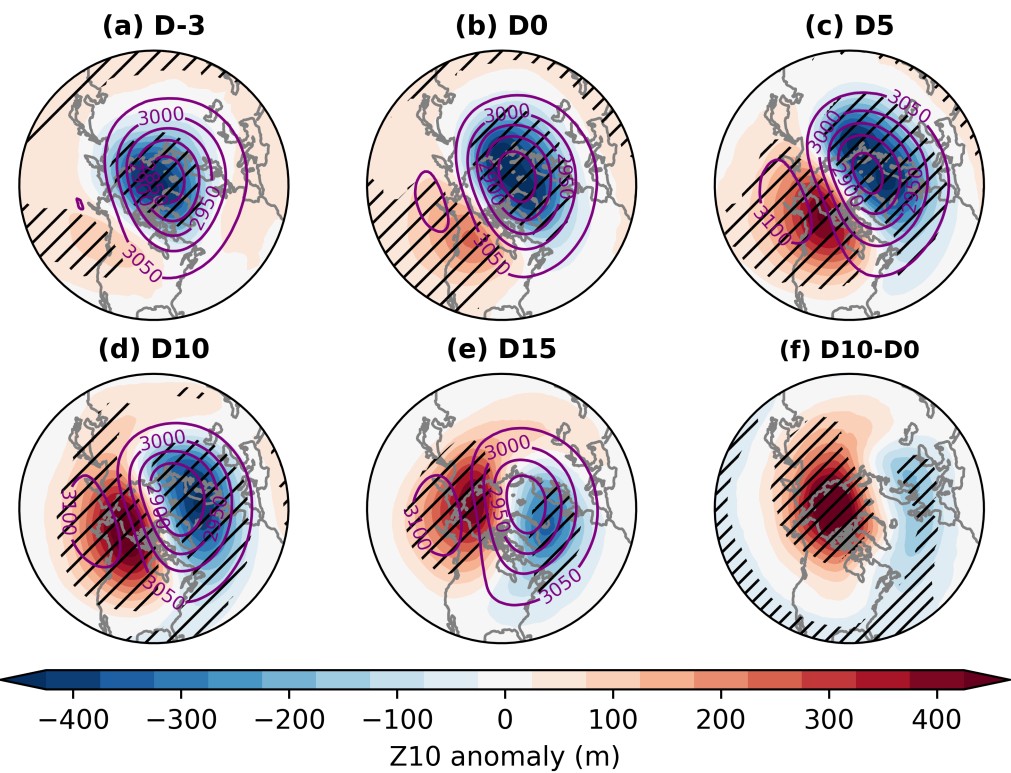

**Figure A4.** (a—e) Composite-mean 10 hPa geopotential height (Z10, contours, dam) and Z10 anomalies (shading, m) at various lags relative to the reflection event onset. (f) Average difference between the Z10 anomalies on day 10 and day 0 (i.e., (d)-(b)). Hatching denotes statistically significant anomalies, assessed as described in Sect. 2.

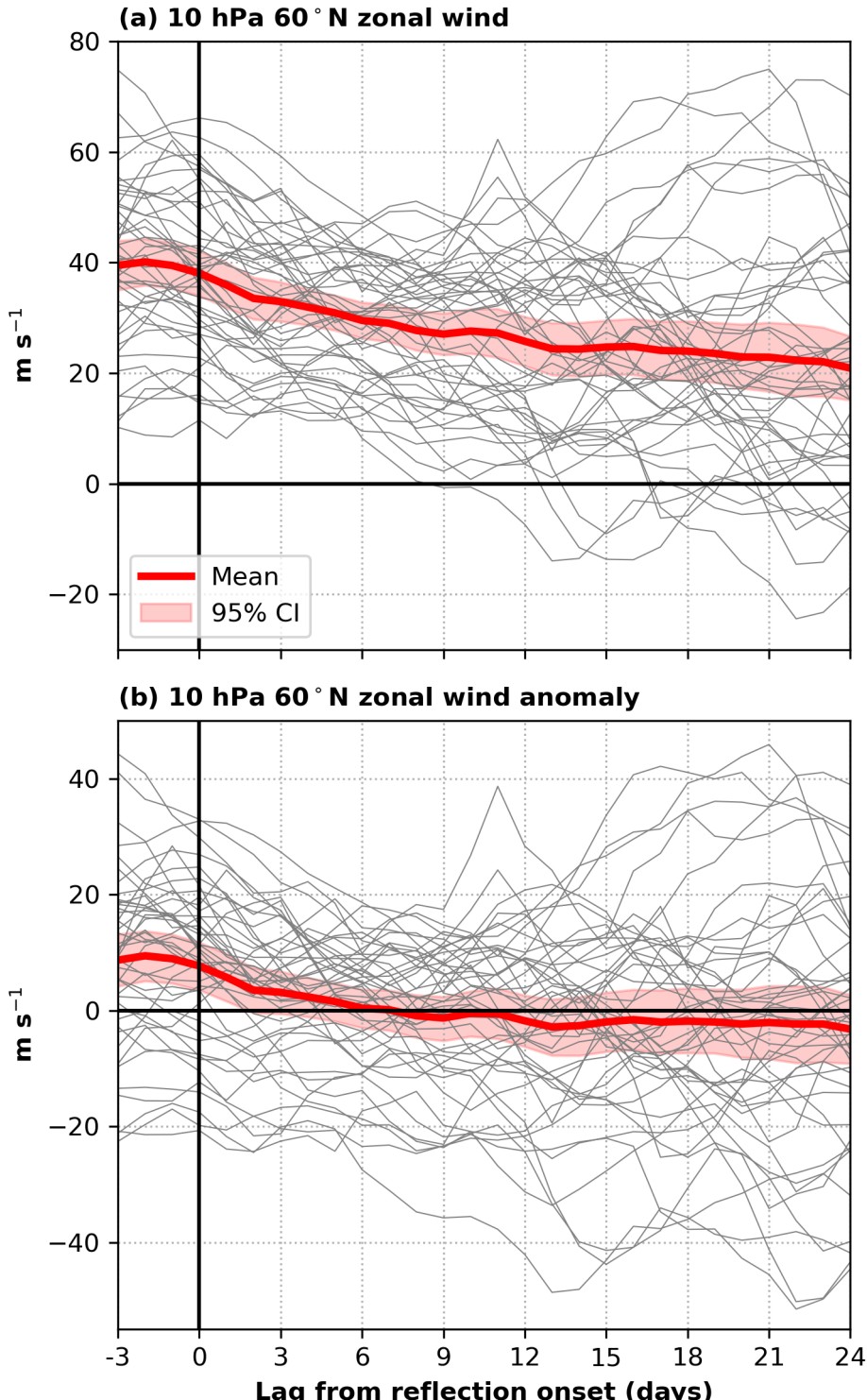

**Figure A5.** Evolution of the 10 hPa 60 °N zonal-mean zonal wind (a) and anomalies (b) for the 44 reflection events. The thick red line denotes the average over all events. Red shading indicates the 95% confidence interval on the mean assessed as described in Sect. 2.

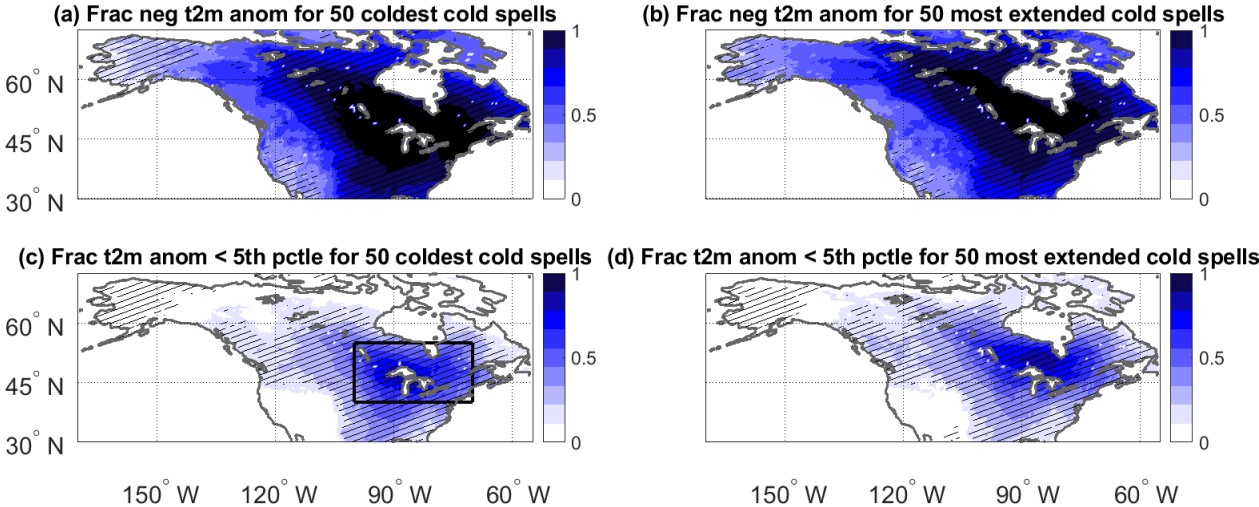

**Figure A6.** Fraction of negative t2m anomalies (a, b) and fraction of t2m anomalies below the local 5th percentile of temperature anomalies (c, d) for the 50 cold spells with the lowest area-averaged temperature anomaly over 40–55 °N and 260–290 °E (a, c; domain shown by the black box) and for the 50 cold spells with the most gidpoints below the local 5th percentile of temperature anomalies over the same domain (b, d). In all cases, a minimum separation of 5 days is enforced between different cold spells. Hatching denotes statistically significant anomalies, assessed as described in Sect. 2.

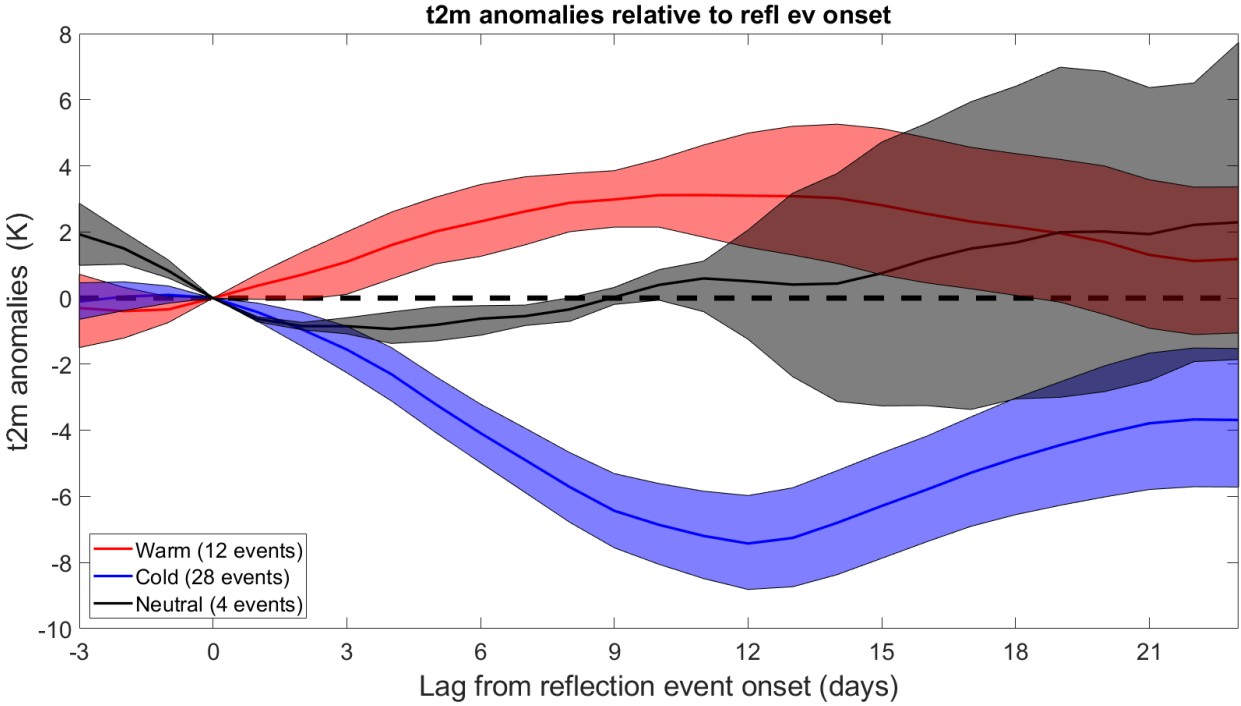

**Figure A7.** Composite-mean t2m anomalies (K) relative to onset of the reflection events for warm (red), cold (blue) and neutral (black) events. These classes of events are defined according to area-averaged t2m anomaly over 40–55 °N and 260–290 °E 10 days after onset: cold events (28) have an anomaly < -0.5 K; warm events (12) have an anomaly > +0.5 K. Events with anomalies between +0.5 and -0.5 K are termed neutral (4). Shading indicates 95% confidence intervals assessed as described in Sect. 2. Due to the small sample size, the confidence interval for neutral events should be interpreted with care.

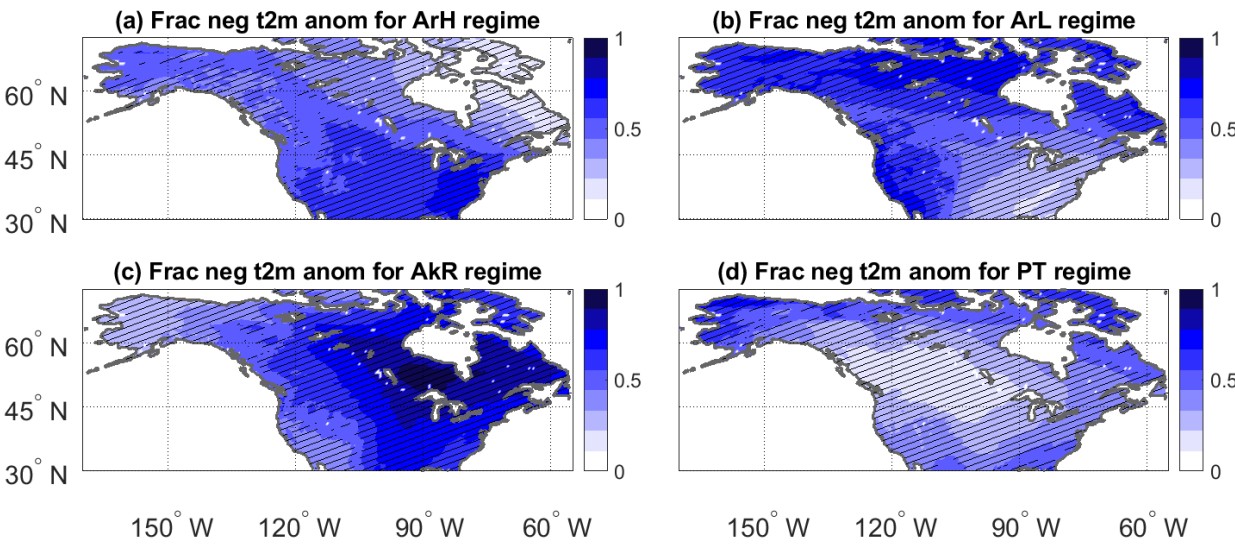

**Figure A8.** Fraction of negative t2m anomalies for days in the: (a) ArH, (b) ArL, (c) AkR and (d) PT weather regimes. Hatching denotes statistically significant anomalies, assessed as described in Sect. 2.

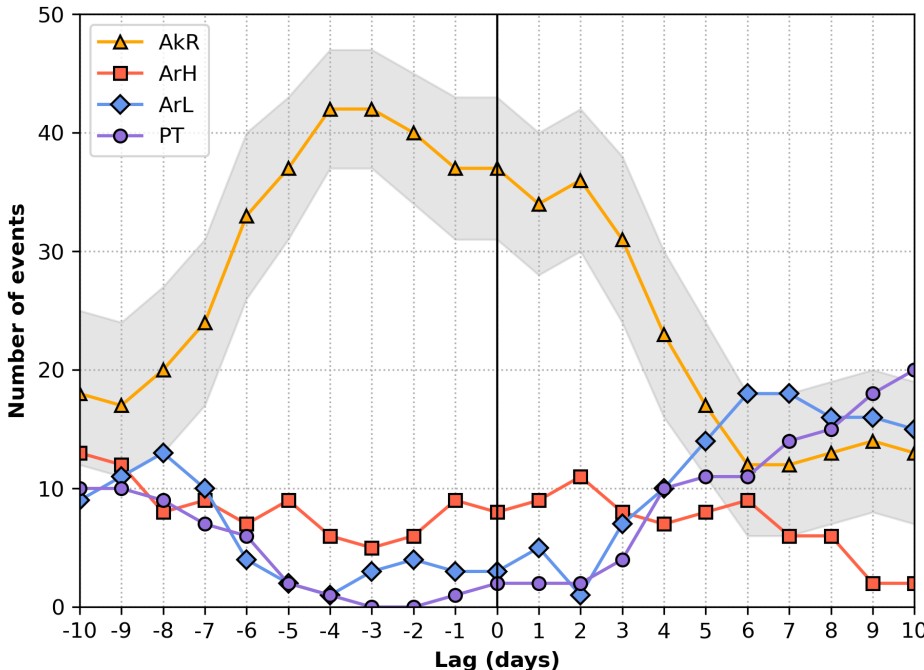

**Figure A9.** Number of weather regime occurrences at different lags centred around the 50 coldest cold spells over 40–55°N and 260–290°E. Note that cold spells within the first 10 days of December and last 10 days of March have been removed. Shading indicates the 95% confidence interval for the AkR regime, assessed as described in Sect. 2.

*Author contributions.*  All authors jointly contributed to designing and writing this study. M.K. and V.W. computed and analysed the reflection index. G.M. conducted the surface temperature analysis. S.H.L. computed and analysed the weather regimes and pressure-level data.

*Competing interests.*  The authors declare that they have no conflict of interests.

*Acknowledgements.*  G.M. received funding from the European Union's H2020 research and innovation programme under ERC grant no. 948309 (CENÆ project). S.H.L. acknowledges funding from NSF grant AGS-1914569 to Columbia University. M.K. has received funding
from the European Union's Horizon 2020 research and innovation program under the Marie Skłodowska-Curie grant agreement (841902). All authors would like to thank P. Zhang and S. Lubis for their input during the review phase, which has helped to develop our analysis.

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
