# Peer review of "Stratospheric Downward Wave Reflection Events Modulate North American Weather Regimes and Cold Spells"

_Weather and Climate Dynamics, 2022_

## Referee Comment (RC2)

Review of manuscript WCD-2022-18: "**Stratospheric Wave Reflection Events Modulate North American Weather Regimes and Cold Spells**", by Messori et al.

This study presents physically interpretable regional impacts of the stratospheric wave reflections on tropospheric circulation anomalies. The main result is that the stratospheric reflection event exhibits a systematic tropospheric evolution from a Pacific Trough regime to an Alaskan Ridge regime, which favors low temperatures over the North America. I find the results interesting and have the potential to trigger other follow-up studies. The manuscript is logically structured and carefully written. However, additional work is required to further clarify the causal relationship between the wave reflection and tropospheric circulation changes, the potential influence of tropospheric internal variability, and other dynamical aspects. Thus, I will recommend major revisions.

**General comments:**

1. My main concern is the true causal relationship between the wave reflection events and the changes in the tropospheric circulation. You showed that the tropospheric imprints of the stratospheric wave reflection are mostly (on averaged) associated with a regime transition from a Pacific Trough (PT) to an Alaskan Ridge (AkR) (e.g., Figure 10). The question here is which comes first. How often stratosphere can be considered a cause of extreme tropospheric events and how often troposphere is considered as the cause of the extreme tropospheric events, which also triggers the stratospheric reflection events. It remains elusive in this paper. As shown in Fig. 10, the PT regime already exists prior to the occurrence of the reflection event and its transition to the AkR regime may not necessarily be related to the reflection event – it could be also triggered by other forcings in the troposphere (e.g., remote forcing from the tropics, blocking etc.). Also, the fact that you see reflection events here could be due to the favorable tropospheric conditions associated with PT, but it does not mean that wave reflection is the cause of the regime transitions or tropospheric changes. Can you address this causality issue? This should be better described and explored.

2. Please keep in mind that the amplitude of the upward propagating waves reflected back into the troposphere decreases with altitude due to the density effect (exp -z/H). Therefore, the wave energy is reduced when it reaches the surface (i.e., damping effect). The question now is how much the energy transferred by the downward planetary wave reflection can explain changes in tropospheric circulation? This could clarify whether the wave reflection event is the cause of the temperature/circulation changes in the troposphere. Another possibility is that the impact of the wave reflection on tropospheric circulation is indirect and requires interaction with synoptic transient eddy forcing via direct effects on baroclinicity and baroclinic eddies (Lubis et al., 2016, 2018; Smy and Scott 2008; Thompson and Birner 2012). Lubis et al., (2016) and Lubis et al., (2018) found the importance of

synoptic-scale eddy-mean flow interaction in shaping the tropospheric response to downward wave reflection event.

- Lubis, S. W., Matthes, K., Omrani, N.-E., Harnik, N. & Wahl, S. Influence of the quasi- biennial oscillation and sea surface temperature variability on downward wave coupling in the northern hemisphere. *J. Atmos. Sci.*, 73, 1943–1965 (2016).

- Lubis, S. W., Matthes, K., Harnik, N., Omrani, N., and Wahl, S. Downward Wave Coupling between the Stratosphere and Troposphere under Future Anthropogenic Climate Change. *Journal of Climate* 31, 10, 4135-4155 (2018).

- Smy, L., & Scott, R. The influence of stratospheric potential vorticity on baroclinic instability. *Quart. J. Roy. Meteor. Soc.*, 135, 1637–1683 (2009).

- Thompson, D. W. J. & Birner, T. On the linkages between the tropospheric isentropic slope and eddy fluxes of heat during Northern Hemisphere winter, *J. Atmos. Sci.*, 69, 1811-1823 (2012).

3. It is well established that negative vertical wind shear results in a reflective layer, with a refractive index $n^2 = 0$ (Harnik and Lindzen, 2001). However, in some cases this condition may also involve a reversal of the zonal-mean zonal wind, with a refractive index $n^2 = \infty$, and a wave absorption (e.g., Kodera et al., 2016). Thus, the downward pointing wave activity vectors (*v'T'* *<0*) in the presence of critical layers does not necessarily mean a linear reflection, if conditions are not linear (Plumb, 2010). For example, the reflective SSW events can be viewed as a mixture between reflective SSW and linear reflection events (e.g., Kodera et al., 2016). The reflective SSW events may involve over-reflection rather than linear reflection because a critical surface formed is embedded by two opposite signs of PV gradient, thus they should be studied separately (Harnik and Heifetz, 2007). My questions are:
- Did you separate the reflective events from the SSW events?
- From the total 44 events, how many reflection events are preceded or followed by the SSW events?
- If you exclude these events, will you have the same temperature responses to a reflection event?
- Especially in Fig. 6(f), the cold anomalies extend further south and are reminiscent of the impact of SSW events. Just make sure that this response does not mix with the SSW events. Please clarify.

Harnik N. and E. Heifetz, 2007. Relating Over-Reflection and Wave Geometry to the Counter Propagating Rossby Wave Perspective: Toward a Deeper Mechanistic Understanding of Shear Instability. J. Atmos. Sci.64 2238-61.

4. Given the fact that the downward stratospheric wave reflection has a significant impact on the cold spell events, it is important to understand the role of natural and

anthropogenic forcings in controlling the variability of such events. Previous studies showed that the natural and anthropogenic forcing (including ENSO, QBO, GHG and ODSs) can significantly influence large-scale circulation that favors downward wave reflection events and the associated surface impacts (e.g., Lubis et al., 2016; Lubis et al., 2018). This should be discussed in the introduction or in the discussion part.

5. The use of v'T' anomaly, instead of total field, as a measure of vertical wave propagation should be used with caution. Especially in Fig. 1a, the region over the North America (inside the polygon) is not only dominated by the climatological mean of negative eddy heat flux. This means a negative RI value does not always indicate downward wave propagation, but a weakening of upward wave propagation. Could you check if the results are robust if you use the total field v'T'? Also check the sensitivity of the results with respect to the size of the regional boxes.

6. What are the dynamical explanations behind the too persistent (prolonged periods) of wave reflection events (Fig. 3)? What causes the negative wind curvature (shear) to persist for such a longer period? Also, the formation of reflective layers does not guarantee that you will always have a reflection event. Without a narrow meridional wave guide channeling the upward wave flux to the reflecting surface, the upward wave propagation most likely disperses in the meridional direction. To answer this question, you most likely need to analyze the wavegeometry ($m^2$ and $l^2$) of those 44 reflection events (see in Perlwitz and Harnik 2003, Shaw et al., 2010, and Lubis et al., 2016).

**Minor comments:**

 Title: I would suggest slightly modifying the title as "Stratospheric **downward** wave reflection events…... ". The *reflection* can be either in the vertical or horizontal direction.

L24-26: Planetary wave patterns are also large-scale meteorological patterns. Please modify this.

L50: It would help the readers to understand what factors controlling the downward wave reflection events. See general comment #4.

L52: *"One potential reason is the difficulty in diagnosing reflection events (see the discussion in Matthias and Kretschmer, 2020)."* I don't think this is the main reason. I think the main reason is the underlying dynamics that are still unclear, especially from the perspective of eddy-mean flow interaction (e.g., baroclinic eddy feedback in the presence of the downward reflection) and energy transfer (e.g., Harnik 2009, Lubis et al., 2017).

L55-59: "*Matthias and Kretschmer (2020) introduced a simple index to identify wave reflection events based on anomalous lower-stratospheric poleward eddy-heat flux over Siberia and Canada*". Matthias and Kretschmer (2020) in fact used the same measure of downward WAF as originally used by Kodera's back in 2008 (see **Figure 2a** in Kodera et al., (2008), with the same level (i.e., 100 hPa) and the same regional locations associated with the upward and downward WAF). It would be nice to acknowledge the work of Kodera et al., (2008) too here.

- Kodera, K., Mukougawa, H., and Itoh, S. (2008), Tropospheric impact of reflected planetary waves from the stratosphere, Geophys. Res. Lett., 35, L16806, doi:10.1029/2008GL034575.

L74: Why October and November are excluded from the analysis? It would be great to argue this from dynamical perspective of downward reflection events (see the seasonality of DWC in Shaw et al., 2010, Lubis et al., 2017 etc).

L102: I would specifically mention that your work only focus on the impacts over the North America sector.

L111: Can you check the sensitivity with respect to the size of the regional boxes used to define the index.

L128: " *..while negative values indicate downward propagation*." Add the following sentence at the end of this "(assuming the wave-activity density is positive definite)".

L134: It may be not true for the whole domain over North America since you are using the anomalous eddy heat flux. See my general comment #5.

L147: "*Finally, following Matthias and Kretschmer (2020) we apply a persistence criterion of 10 days*". Would be the results sensitive with the choice of this threshold? Please clarify.

L164: "*we analyze the tropospheric evolution associated with the 44 reflection events*". How many events are related to SSW event? Please see my general comment #3.

Figure 6. Can you explain dynamically how does the wave reflection event cause a stretching tropospheric jet, hence a severe cold spell over North America? Can you quantify this? I think the elongation of tropospheric jet (i.e., wavier) is of importance to the cold spell over North America, which may provide useful insight to the forecast of cold spells over North America.

L260-L265: I am afraid that your signals are mixed with the SSW events. Please do check. See my general comment #3.

L321-L323: Do you have any cases where a downward reflection event is not preceded by the PT regime, but still have a strong influence on surface temperature? Or a case when you have reflection but there is no significant impact on the surface temperature. I am wondering whether the PT regime is a robust tropospheric precursor of the downward reflection event.

L325: Very good point but without exploring this, it remains elusive if wave reflection events are the main cause of the cold spells.

.

---

## Author Response (AR1)

**These replies match the ones posted in the public discussion. Where appropriate, we have made edits to figure numbers or textual quotes to ensure consistency with the revised manuscript.**

**We have additionally corrected the following typo: Reviewer #1, Specific comment #6, 21 days → 28 days**

**Reviewer #1**

This manuscript analyzes the stratospheric wave reflection events and the following tropospheric/surface weather evolutions in North America using reanalysis data. The authors start with a regional stratospheric wave reflection detection metric and 44 events have been detected since 1980. By comparing with the North American weather regimes, the wave reflection events show a transition from a regime characterized as a trough anomaly in North Pacific and a ridge anomaly in North America to a regime with a reversed pattern. The surface air temperature changes from warm anomaly to cold anomaly in about two weeks, presenting a large-scale decrease evolution.

I think the two mechanisms are in fact different aspects of the same planetary wave-zonal flow interaction rather than totally different dynamical mechanisms. Both wave-mean flow interaction and wave reflection are useful for our understanding of the influence of the stratosphere on the troposphere. While the wave-mean flow interaction has been well studied, relatively less attention has been paid to wave reflection. The current study is a systematic analysis of the tropospheric weather evolution following the detected wave reflections. The manuscript is straightforward and well-organized. I believe this manuscript could be a helpful reference for the research community of stratosphere-troposphere coupling. Thus, I think it could be acceptable after minor revision.

Thank you for your positive outlook on our submission and suggestions for further improvement. We agree that both wave-mean flow interactions and wave reflection are both a form of wave-flow interaction, and have added a brief comment to that effect in the introduction of our manuscript. We have copied your comments below in *Italics*, with our replies in Blue.

**Specific comments:**

*1. What's the spatial pattern of the stratospheric polar vortex associated with the detected wave reflection events? Stretching, shifting, or zonally symmetric weakening? The authors may add an Appendix figure at least.*

We thank the reviewer for this suggestion. In the revised manuscript we have included the new Fig. A4 (also copied below) showing the mean evolution of Z10 for the same lags as the Z500 anomalies. This shows an anomalously strong vortex (negative Z10 anomalies over the central Arctic) on average during the reflection event. This is consistent with the expected favorable condition for wave reflection. At the onset of the reflection event, there is

a small signature of an Aleutian anticyclone, which intensifies at positive lags. This is somewhat associated with stretching the vortex along 90°W by D10-15 (reminiscent of Fig. 2 in Cohen *et al.* 2021), although not notably so. To enable a better comparison with Cohen *et al.* 2021, we also include below the Z100 composites as FIg. R1. We now briefly comment on this at the end of Sect. 3 and in Sect. 6.

Judah Cohen, Laurie Agel, Mathew Barlow, Chaim I Garfinkel, and Ian White. Linking Arctic variability and change with extreme winter weather in the United States. *Science*, 373(6559):1116–1121, 2021

[Figure]

FIg. A4: *(a–e) Composite-mean 10 hPa geopotential height (Z10, contours, dam) and Z10 anomalies (shading, m) at various lags relative to the reflection event onset. (f) Average difference between the Z10 anomalies on day 10 and day 0 (i.e., (d)-(b)). Hatching denotes statistically significant anomalies, assessed as described in Sect. 2 in the main text.*

[Figure]

*Fig. R1: (a–e) Composite-mean 100 hPa geopotential height (Z100, contours, dam) and Z100 anomalies (shading, m) at various lags relative to the reflection event onset. (f) Average difference between the Z100 anomalies on day 10 and day 0 (i.e., (d)-(b)). The 1580 dam contour is bolded. Hatching denotes statistically significant anomalies, assessed as described in Sect. 2 in the main text.*

*2. Line 115-117: I'm wondering why the authors employed a lower threshold (1.5->1). To allow more samples? I'd guess the conclusions still hold with different thresholds, such as 1.5 or 2, but the authors may double-check and make a clear statement.*

The threshold of 1 was chosen here because the associated events already fulfill the assessed criteria (as shown in Fig. 1 and Fig. 2). We have tested using higher thresholds of RI > 1.5 and RI > 2, which naturally reduces our sample size (35 and 27 events respectively). The qualitative results concerning the surface anomalies associated with the reflection events are nonetheless consistent with those obtained for RI > 1. We further note that while there are some regions in which the temperature anomalies for the higher RI thresholds are larger than those for RI > 1, the difference is generally small. This points to RI>1 being a threshold which selects reflection events that are sufficiently intense to be associated with a large surface impact. We show in Figs. R2–R5 below the results corresponding to Figs. 4c, d and Fig. 5 in the main text. We now mention this in the revised text in Sect. 3.

[Figure]

*Fig. R2. Same as Fig. 4c, d in the main paper, but for RI > 1.5 (35 Events).*

[Figure]

*Fig. R3. Same as Fig. 4c, d in the main paper, but for RI > 2 (27 Events).*

[Figure]

*Fig. R4. Same as Fig. 5 in the main paper, but for RI > 1.5.*

[Figure]

*Fig. R5. Same as Fig. 5 in the main paper, but for RI > 2.*

3. Table A1: event#42 the event detected in the 2017-2018 winter is different from their previous study (MK20). The authors may explain why.

MK20 used the term "events" to refer to three mid-latitude cold spells that occurred in the winter 2017/18. In contrast, we use it here to refer to the occurrence of wave reflection. Fig. 10 in MK20 shows the evolution of the reflection index in that winter (red line) which shows that the index was high (>1.5) during the second half of January, which is consistent with event#42 from this study (13. Jan 2018 - 2 Feb 2018). The reflection event preceding the cold spell at the end of December 2017 ("event 1" in MK20) is indeed not included in this

study as its start date is in November which we excluded here (see also our answer to the next comment). We have now clarified this point in Sect. 2 of the main text.

*4. Table A1: Is it possible for a reflection event to occur in November?*

It is indeed possible (see our answer to the previous comment), but we decided to focus on DJFM only, primarily because it is the period with the largest stratosphere-troposphere coupled variability (e.g. Fig 2a in Baldwin *et al.* 2003). Furthermore, the regimes analysis was previously defined only for DJFM (Lee *et al.* 2019, 2022), and cold extremes are typically colder during this period due to the seasonal cycle of temperature. We also note that it is often DJFM which is considered 'extended winter' for analysis of wintertime climate modes (e.g. https://climatedataguide.ucar.edu/climate-data/hurrell-wintertime-slp-based-northern-annular -mode-nam-index). In the revised manuscript we nonetheless now mention the possibility of the occurrence of wave reflection events in October and November not covered by our study in Sect. 2.

SH Lee, JC Furtado, and AJ Charlton-Perez. Wintertime North American weather regimes and the Arctic stratospheric polar vortex. *Geophysical Research Letters*, 46(24):14892–14900, 2019

Simon H Lee, Andrew J Charlton-Perez, Steven J Woolnough, and Jason C Furtado. How do stratospheric perturbations influence North American weather regime predictions? *Journal of Climate*, 2022. https://doi.org/10.1175/JCLI-D-21-0413.1

*5. Table A1: It would be helpful for readers if the authors can add a column to show whether the event is associated with a major SSW.*

Thank you for this very good suggestion. We have added this information to the table, basing it on NOAAs Sudden Stratospheric Warming Compendium data set (https://csl.noaa.gov/groups/csl8/sswcompendium/majorevents.html) with the addition of the January 2021 SSW described in Lee (2021). An SSW occurs during 9 of our 44 events; however, only 3 of the SSW events occur within 15 days of the onset of a reflection event, which is the period our analysis focuses on. None occur within 9 days. The total number of SSWs considered is 27. We have further added some discussion of the correspondence or lack thereof between our reflective events and SSWs to the text in Sect. 6, also in reply to several of the comments of Reviewer #2.

Baldwin, M. P., Stephenson, D. B., Thompson, D. W., Dunkerton, T. J., Charlton, A. J., & O'Neill, A. (2003). Stratospheric memory and skill of extended-range weather forecasts. *Science*, *301*(5633), 636-640.

Lee, S.H. (2021), The January 2021 sudden stratospheric warming. *Weather*, **76**: 135-136. https://doi.org/10.1002/wea.3966

6. Line 133 "except for a couple of days": I don't expect much difference, but just wondering whether the authors removed these cases in the following analysis, given they are not strictly downward? If a higher threshold (e.g., RI>1.5) is used, is it possible (v't')Canada is completely lower than 0?

During our 44 reflection events (defined as in the paper with a threshold of RI>1.0 and minimum 10 days in duration; total number of reflection event days: 1012), there are in total 28 days for which v'T' averaged over the Canadian sector is greater than or equal to zero. This corresponds to only 2.7% of all days during reflection events. When using a threshold of RI>1.5 the corresponding value is 9 days. Since the fraction of positive v'T' over the Canadian sector is very small for both thresholds, we do not expect this to affect our results – as also supported by the similarity between the surface anomalies for RI > 1.5 and for RI > 1 shown in Figs. R2–R5.

Due to the small fraction of v'T' reflective days (RI > 1) showing positive values over the Canadian box, on average during reflective days almost all grid-points in this region show negative values (see Figure R6a below). The few days with domain-averaged positive v'T' values still display regions of negative values, yet these are limited in their spatial extent. Figure R6b shows an example of a reflective event with positive average v'T' over Canada.

[Figure]

*Fig. R6: a) Average over all reflection days (RI>1.0); b) example of an event with domain-averaged positive v'T' over Canada on 12.02.1980.*

We believe that these very few cases have a negligible effect on our analysis. In the revised version of the manuscript we now state the exact number of cases when v'T' over Canada is positive.

**Technical comments:**

*1. Line 45: Zhang et al. 2020 discussed the North American cold spells following stratospheric anomalies mainly through the lens of wave reflection by giving the pre-existing Arctic surface conditions, which is more relevant to the discussion in Line 47, 50, and 320.*

We appreciate that in Zhang *et al.*, 2020, the authors discuss both the regional effect of planetary waves and the role of vertical propagation of wave anomalies in engendering North American cold spells. However, their analysis is framed around the role of SSWs, which we found have only a minimal link to wave reflection events leading to surface cold spells (only 4 of the "cold" reflection events correspond to a SSW). Moreover, there is little resemblance between the mid-tropospheric anomalies shown in Fig. 4 in Zhang *et al.*, and the anomalies we observe in our Fig. 7. We do nonetheless agree that the point Zhang *et al.* make concerning the role of surface pre-conditioning is highly relevant to the discussion of the origin of the mechanistic chain of events in Sect. 6. We now refer to Zhang *et al.*, (2020) in that paragraph. We have additionally modified the passage where Zhang *et al.* (2020) are cited in the introduction to acknowledge the reflection-like mechanisms discussed in that study.

2. Line 123: It would be better to state the term1 in Eq.1 (v't')sib >0?

As suggested, we have added the respective mathematical formulations to our list.

3. Line 124: similar comment here, (v't')can <0?

As suggested, we have added the respective mathematical formulations to our list.

*4. Line 221 "Fig. A1": This is the first time Fig. A1 is called while Fig. A2-A4 are called earlier. The authors may adjust the order of the Appendix figures.*

We mistakenly omitted referring to Fig. A1 in Sect. 2. We have now added this reference and the Appendix figures should appear in the order they are referred to. Note that due to new analysis performed during the revision the numbering of figures in the Appendix has changed.

*5. Line 139 "vertical wind shear": Do the authors mean "vertical shear of zonal wind"?*

Yes, that is indeed what we meant, and have now corrected it in the text both here and at a later instance.

*6. Line 189 "the southernmost portion of the USA": Although this is not a piece of key information, one cannot see that in the figures. The authors may slightly enlarge the domain.*

This was bad phrasing on our part. In Fig. 4d, which is the panel we are commenting on in that passage, there is actually not much signal south of the chosen domain (see Fig. R7 below showing a domain extended by 10 degrees to the South). What we meant to point to was actually the bottom right-hand corner of the shown domain. We have now edited this in the text: "Only Alaska and the south-eastern corner of the domain show neutral or weakly positive anomalies (Fig. 4d)."

[Figure]

*Fig. R7: Same as Fig. 4d, but for an extended meridional domain (20–70 °N).*

*7. Line 227-228 "using k-mean clustering of the leading 12 PCs of the daily Z500 anomalies": I'm not sure my understanding is correct. Did the authors reconstruct new Z500 daily anomalies using the leading 12 EOFs and their time series, and then assign this new reconstructed daily field to 4 groups using clustering?*

We followed a relatively standard method whereby we first computed the timeseries of the leading 12 PCs of the daily Z500 anomalies, then performed k-means clustering of these PCs to obtain the centroid coordinates in PC space, and then assigned each day to a 'regime' by determining the nearest cluster centroid by Euclidean distance in 12-dimensional PC space. We have added some clarifying discussion, although we emphasize that the method here is by no means novel. Some more detailed discussion can be found in Robertson et al. (2020), which we cite in the main text.

Robertson, A. W., Vigaud, N., Yuan, J., & Tippett, M. K. (2020). Toward identifying subseasonal forecasts of opportunity using North American weather regimes. *Monthly Weather Review*, *148*(5), 1861-1875.

*8. Line 236-237 "before … than climatology": Are the authors talking about the evolution from day 0 to day 5  and then to day 8 in Fig.8c?*

We have rephrased this passage, which now reads: " The AkR regime is unlikely immediately prior to the onset of the reflection event. It then approximately triples in frequency within the first five days after the event onset to become slightly more frequent than climatology and peaks in frequency around days 9 to 12. At all subsequent positive lags shown here, the regime is more frequent than in the days before the event onset (Fig. 8c)"

*9. Figure A4: the authors may show the number of cases used to calculate each line in the figure legend.*

We have added these numbers to the legend as suggested.

**Reviewer #2**

This study presents physically interpretable regional impacts of the stratospheric wave reflections on tropospheric circulation anomalies. The main result is that the stratospheric reflection event exhibits a systematic tropospheric evolution from a Pacific Trough regime to an Alaskan Ridge regime, which favors low temperatures over the North America. I find the results interesting and have the potential to trigger other follow-up studies. The manuscript is logically structured and carefully written.

However, additional work is required to further clarify the causal relationship between the wave reflection and tropospheric circulation changes, the potential influence of tropospheric internal variability, and other dynamical aspects. Thus, I will recommend major revisions.

We thank the Reviewer for the critical yet constructive outlook on our submission and suggestions for further improvement. We have performed significant additional analysis on the dynamical aspects of wave reflection in response to the points raised by the Reviewer. We have copied the comments below in *Italics*, with our replies in Blue.

**General comments:**

1. My main concern is the true causal relationship between the wave reflection events and the changes in the tropospheric circulation. You showed that the tropospheric imprints of the stratospheric wave reflection are mostly (on averaged) associated with a regime transition from a Pacific Trough (PT) to an Alaskan Ridge (AkR) (e.g., Figure 10). The question here is which comes first. How often stratosphere can be considered a cause of extreme tropospheric events and how often troposphere is considered as the cause of the extreme tropospheric events, which also triggers the stratospheric reflection events. It remains elusive in this paper. As shown in Fig. 10, the PT regime already exists prior to the occurrence of the reflection event and its transition to the AkR regime may not necessarily be related to the reflection event – it could be also triggered by other forcings in the troposphere (e.g., remote forcing from the tropics, blocking etc.). Also, the fact that you see reflection events here could be due to the favorable tropospheric conditions associated with PT, but it does not mean that wave reflection is the cause of the regime transitions or tropospheric changes. Can you address this causality issue? This should be better described and explored.

To address this comment we have conducted both an analysis of the eddy height field, which can provide insights into the dynamics of the reflection events, and a statistical analysis of the North American weather regime transitions.

The eddy height field during reflection events (new Fig. 12 below, also included in the revised manuscript) shows a tropospheric pattern associated with the PT regime, which is linked to a westward-tilting/upward propagating wave packet over Siberia. In other words, the analysis suggests that a pattern similar to PT is favorable for the enhanced upward propagation from Siberia, implying that some aspect of the causal chain leading to stratospheric reflection events indeed originates in the troposphere. We now comment on this figure in Sect. 6.

However, we also note that PT is the most frequent regime (31.6% of DJFM days), and thus *far* more common than reflection episodes – which is reminiscent of the relationship between blocking and SSWs (blocking being far more common). This suggests a key role of the stratosphere in guiding the waves downward, further evidenced by the structure of the wave field showing an eastward tilting wave structure over the Canada node.

To further investigate the statistics of the PT to AkR transition, we have constructed a weather regime transition matrix (Fig. 9 below). Largely due to the inherent persistence of the regimes, the bulk of the transitions are from a regime to itself (lag-1 persistence, shown along the forward-sloping diagonal in Fig. 9a), and the number of samples for each individual transition pathway is small. For PT, the transition to AkR is the least likely (4.4%) and the fourth least-likely of all 12 possible regime transitions, whilst PT is itself also the most persistent (84.1%). The low number of  PT–AkR transitions is consistent with Lee et al. (2022), who show that the centroids of PT and AkR are almost equal-and-opposite in both of the leading two EOFs, thus requiring a particularly large change to the dominant flow patterns in order to instigate the change. This can be contrasted with the ArH–PT transition, which is the most likely, and consistent with the centroids of ArH and PT effectively overlapping in EOF1. The number of PT–AkR transitions in our dataset – 70 episodes – is of the same order of magnitude as the number of reflection events (44). To further support this link, in Fig. 9b we show for each transition pathway the percentage of those transitions where the D0 RI is greater than 1 (i.e., the percent of transitions associated with wave reflection). Bold indicates statistical significance at the 5% one-sided level. Only the proportion of transitions to AkR associated with wave reflection are significantly greater than what might occur by random sampling. While this does not indicate the direction of causality, it does confirm a close connection between the PT–AkR transition and the reflection events. We have included this figure and a discussion of it in Sect. 5 in the main text.

[Figure]

Figure 12: Evolution of the mean vertical wave structure associated with reflection events. Black contours denote the average 40-80°N eddy geopotential height field (contours every 100m, dashed negative, zero contour thickened). Shading denotes the departure of the eddy height field from climatology. Stippling denotes anomalies significantly different from zero, , assessed as described in Sect. 2 in the main text. Pink vertical lines delineate the

*longitudinal range of the Siberian box (140-200°E), and green vertical lines delineate the longitudinal range of the Canadian box (230-280°E).*

[Figure]

*Figure 9: (a) Transition matrix for the four North American weather regimes. For each initial regime (x-axis), the numbers in each column denote the observed probability (expressed as a percentage; columns sum to 100) of persisting in the same regime (white font) or transitioning into a different regime. The total number of instances of each transition (N) is also shown. (b) Proportion of each transition pathway which occurs with RI>1 on the day prior to the transition (D0), expressed as percentages. p-values indicate the estimated probability of obtaining a statistic greater than the observed value by chance, obtained by randomly re-sampling all DJFM days 10,000 times (without replacement) using the observed sample sizes for each transition pathway. Where this is less than 0.05 (i.e., one-sided 5% significance), the observed statistic is shown in bold white font.*

To conclude, we agree that attributing causality between tropospheric reflection events and tropospheric circulation regimes is complex, especially because of the association between tropospheric precursors and stratospheric events. This is a difficulty that is not specific to wave reflection, and indeed also holds true for SSWs – and more broadly for stratosphere-troposphere interactions. We nonetheless believe the new eddy height field and weather regime transition analyses provide some further clues on this issue in the context of our work. Finally, we believe there is some value in our results beyond a full causality analysis, as we clearly demonstrate that reflection events as defined by our index on average precede North American cold spells, indicating their predictive potential for such high-impact extreme events.

*2. Please keep in mind that the amplitude of the upward propagating waves reflected back into the troposphere decreases with altitude due to the density effect (exp -z/H). Therefore, the wave energy is reduced when it reaches the surface (i.e., damping effect). The question now is how much the energy transferred by the downward planetary wave reflection can explain changes in tropospheric circulation? This could clarify whether the wave reflection event is the cause of the temperature/circulation changes in the troposphere. Another*

*possibility is that the impact of the wave reflection on tropospheric circulation is indirect and requires interaction with synoptic transient eddy forcing via direct effects on baroclinicity and baroclinic eddies (Lubis et al., 2016, 2018; Smy and Scott 2008; Thompson and Birner 2012). Lubis et al., (2016) and Lubis et al., (2018) found the importance of 2 synoptic-scale eddy-mean flow interaction in shaping the tropospheric response to downward wave reflection event.*

*• Lubis, S. W., Matthes, K., Omrani, N.-E., Harnik, N. & Wahl, S. Influence of the quasi-biennial oscillation and sea surface temperature variability on downward wave coupling in the northern hemisphere. J. Atmos. Sci., 73, 1943–1965 (2016).*
*• Lubis, S. W., Matthes, K., Harnik, N., Omrani, N., and Wahl, S. Downward Wave Coupling between the Stratosphere and Troposphere under Future Anthropogenic Climate Change. Journal of Climate 31, 10, 4135-4155 (2018).*
*• Smy, L., & Scott, R. The influence of stratospheric potential vorticity on baroclinic instability. Quart. J. Roy. Meteor. Soc., 135, 1637–1683 (2009).*
*• Thompson, D. W. J. & Birner, T. On the linkages between the tropospheric isentropic slope and eddy fluxes of heat during Northern Hemisphere winter, J. Atmos. Sci., 69, 1811-1823 (2012).*

The notion of stratospheric wave reflection and the fact that it can have impacts on the troposphere is relatively well-established in the literature, although the details of the dynamical processes involved are not yet fully understood, as also pointed out by the Reviewer in one of their comments below. In this study, we do not seek to provide a dynamical analysis of the wave reflection process, which as the Reviewer notes, has attracted considerable attention in the past literature, and indeed the energetics of the wave reflection is not an aspect we consider anywhere in our analysis. The Reviewer's suggestion, while interesting, does not reference any specific aspect of our submission and would likely require one or more entirely new studies. We nonetheless agree that it is worth discussing in our text the possible existence of a direct or indirect pathway for the impact of wave reflection on tropospheric processes. We have added some considerations to this effect in Sect. 6 of the main text.

*3. It is well established that negative vertical wind shear results in a reflective layer, with a refractive index $n2 = 0$ (Harnik and Lindzen, 2001). However, in some cases this condition may also involve a reversal of the zonal-mean zonal wind, with a refractive index $n2 = \infty$, and a wave absorption (e.g., Kodera et al., 2016). Thus, the downward pointing wave activity vectors (v'T' <0) in the presence of critical layers does not necessarily mean a linear reflection, if conditions are not linear (Plumb, 2010). For example, the reflective SSW events can be viewed as a mixture between reflective SSW and linear reflection events (e.g., Kodera et al., 2016). The reflective SSW events may involve over-reflection rather than linear reflection because a critical surface formed is embedded by two opposite signs of PV gradient, thus they should be studied separately (Harnik and Heifetz, 2007). My questions are:*
*• Did you separate the reflective events from the SSW events?*
*• From the total 44 events, how many reflection events are preceded or followed by the SSW events?*

*• If you exclude these events, will you have the same temperature responses to a reflection event?*
*• Especially in Fig. 6(f), the cold anomalies extend further south and are reminiscent of the impact of SSW events. Just make sure that this response does not mix with the SSW events. Please clarify.*

*Harnik N. and E. Heifetz, 2007. Relating Over-Reflection and Wave Geometry to the Counter Propagating Rossby Wave Perspective: Toward a Deeper Mechanistic Understanding of Shear Instability. J. Atmos. Sci.64 2238-61.*

The point about the need to take into account the possible presence of SSWs leading to different reflection dynamics is well made. While the U2-10 reflection index from Perlwitz & Harnik indeed shows negative values during SSWs, that is, indicating negative vertical wind shear (making it not fully suitable to differentiate between reflection events and SSWs), this is usually not the case for our regional reflection index RI, as also discussed in more detail in MK20.

To show this, we have now verified the relationship betweenf our reflective events and SSWs, using the Sudden Stratospheric Warming Compendium data set (https://csl.noaa.gov/groups/csl8/sswcompendium/majorevents.html) and the Jan 2021 SSW described in Lee (2021). The total number of SSWs included in these sources is 27. Considering the *full duration* of the reflective events, an SSW occurs during 9 of our 44 events. In only 3 events does an SSW occur within the 15 days following the reflection event onset, which is the period we focus on in our analysis, and so SSWs do not dilute our results. Indeed, the onset of reflection events typically corresponds to a strengthened stratospheric polar vortex (see Figs. A4 and R1 in the replies to Reviewer #1). This information has been included in Table A1. 5 of the 9 events during which an SSW occurs are not "cold" reflective events. In other words, they are part of the minority of reflective events not associated with a strong surface cooling over the North American continent. This is consistent with the fact that SSWs are not typically associated with *severe* cold weather over central North America on synoptic to super-synoptic timescales (relative to the onset of the 'events' considered here).

To further demonstrate that our reflection events do not occur following U10-60<0 conditions, which would have made it difficult to discern whether they were reflection events or SSWs, we show in Fig. R8 below the U10-60N evolution for reflective events and find only *three* major SSWs in the period we primarily analyse for our reflection events (i.e. lags out to 15 days). These are the reflection events of 10 Jan 1987 (SSW on 23 Jan 1987), 27 Dec 2003 (SSW on 5 Jan 2004) and 24 Dec 2012 (SSW on 7 Jan 2013). We see this as further evidence that our results are largely unaffected by major SSWs, which are related to wave absorption rather than wave reflection. We also observe in Fig. R8b that the U10 covers a large spread of anomalies, confirming that wave reflection is a form of stratospheric variability largely distinct from the vortex strength/NAM variability.

[Figure]

*Fig. R8: U10-60N evolution for reflective events (a) and corresponding anomalies (b) at several lags relative to reflection event onset (day 0). The thick red line shows the mean across all events, and shading shows a 95% confidence interval on the mean.*

We have further repeated our temperature composites from Fig. 6f excluding the 9 reflection events associated with SSWs (Fig. R9 below). We find that, as expected from the above discussion, the cold temperature anomaly footprint is somewhat enhanced, especially at longer lags. The negative anomalies in panel (f) in the southern part of our analysis domain

are still present. Indeed, SSW-like stratospheric configurations typically correspond to only modest negative temperature anomalies there (e.g. see Fig. 4b in Kretschmer *et al.*, 2018)

We have added the figure showing the 10 hPa geopotential height anomalies associated with the reflection events to the study as the new Fig. A4. We have further added some discussion of the correspondence or lack thereof between our reflective events and SSW to the text in Sect. 6, including a summary of the important dynamical aspects that the Reviewer has raised in their comment.

[Figure]

*Fig. R9: Same as Fig. 6 in the main paper, but excluding from the composite the 9 reflection events matching SSW as indicated in Table A1.*

Baldwin, M. P., Stephenson, D. B., Thompson, D. W., Dunkerton, T. J., Charlton, A. J., & O'Neill, A. (2003). Stratospheric memory and skill of extended-range weather forecasts. *Science*, *301*(5633), 636-640.

Kretschmer, M., Cohen, J., Matthias, V., Runge, J., & Coumou, D. (2018). The different stratospheric influence on cold-extremes in Eurasia and North America. *npj Climate and Atmospheric Science*, *1*(1), 1-10.

Lee, S.H. (2021), The January 2021 sudden stratospheric warming. *Weather*, **76**: 135-136. https://doi.org/10.1002/wea.3966

*4. Given the fact that the downward stratospheric wave reflection has a significant impact on the cold spell events, it is important to understand the role of natural and anthropogenic forcings in controlling the variability of such events. Previous studies showed that the natural and anthropogenic forcing (including ENSO, QBO, GHG and ODSs) can significantly influence large-scale circulation that favors downward wave reflection events and the*

*associated surface impacts (e.g., Lubis et al., 2016; Lubis et al., 2018). This should be discussed in the introduction or in the discussion part.*

We agree that on longer timescales than those associated with meteorological predictability, controls by large-scale modes of variability and anthropogenic forcing may be highly relevant. Specifically, one can hypothesise that the modulation of reflection events by such forcings could provide a significant control on North American cold spell frequency. We have now added a paragraph discussing this point in Sect. 6. We have additionally investigated the statistical link between the QBO phase and the occurrence of stratospheric wave reflection events, as described in detail in our reply to the comment on l. 50, and find a clear association between reflection events and QBOw. We also briefly discuss this in Sect. 6.

*5. The use of v'T' anomaly, instead of total field, as a measure of vertical wave propagation should be used with caution. Especially in Fig. 1a, the region over the North America (inside the polygon) is not only dominated by the climatological mean of negative eddy heat flux. This means a negative RI value does not always indicate downward wave propagation, but a weakening of upward wave propagation. Could you check if the results are robust if you use the total field v'T'? Also check the sensitivity of the results with respect to the size of the regional boxes.*

We agree that interpreting v'T' anomalies is difficult without knowing the climatology. In Figure 1d and the accompanying discussion in Sect. 3, we show (as in the original text) the *full v'T' field* (not its anomaly)*,* which confirms that the RI>1 days represent both *enhanced upward* and *increased downward* wave propagation. Figure R10 below shows the average total meridional heat flux v'T' during reflection events (similar to Figure 1b in the manuscript but with total fields instead of anomalies). There is only a very small area in the south-west corner of the Canadian box with slightly positive values. Thus, during reflection events the flux in the Canadian sector is negative and even stronger than on average. The domains used to compute the RI (which are slightly adapted compared to MK20) were chosen after performing several sensitivity tests. A more detailed answer on the domain sensitivity can be found below (comment to L 111).

[Figure]

*Fig. R10 Average total meridional heat flux during reflection events.*

6. What are the dynamical explanations behind the too persistent (prolonged periods) of wave reflection events (Fig. 3)? What causes the negative wind curvature (shear) to persist for such a longer period? Also, the formation of reflective layers does not guarantee that you will always have a reflection event. Without a narrow meridional wave guide channeling the upward wave flux to the reflecting surface, the upward wave propagation most likely disperses in the meridional direction. To answer this question, you most likely need to analyze the wave geometry (m2 and l2 ) of those 44 reflection events (see in Perlwitz and Harnik 2003, Shaw et al., 2010, and Lubis et al., 2016).

The Reviewer raises two distinct points here, one on the dynamical aspects of wave reflection, and one on the duration of the reflection events we identify.

Concerning the dynamical aspects of wave reflection, this requires both a reflective layer and a meridional waveguide. The existence of a reflective layer is shown in Figure 2 in the paper. The new Figure A2 (copied below) shows the meridional profiles of the zonal wind at 30 hPa averaged over the Canadian sector. The red line represents the climatology, the blue line the average of the non-reflective days (RI<1) and the green line the average of the reflective days (RI>1). Similar to findings from MK20, there is an increased curvature in polar latitudes and mid latitudes, supportive of the existence of a meridional waveguide during reflection events. This shows that our index identifies events consistent with the dynamics expected of wave reflection. While a detailed wave geometry analysis is especially useful for case studies,  our simple index is thus suitable for event detection in a climatological

perspective. This and related issues are also discussed in more detail in MK20 in Sect. 3b (i) and (ii), where the authors present challenges in diagnosing wave reflection events and an analysis in support of their choice of reflection index. We have now added Figure A2 below to the Appendix of our study, and comment on it and on the importance of a waveguide in Sect. 3.

[Figure]

*Fig. A2: Meridional profiles of the zonal wind averaged between 230°-280°E (Canadian sector) at 30 hPa.*

Regarding the very persistent events, we have identified the reflection events with a duration in excess of 40 days and provide a brief context for these below:

- 12-Dec-1993 (46 days) – there were several vortex vacillations in this period, with a 'minor warming' or 'sudden deceleration' during late December 1993/January 1994.
- 04-Jan-1995 (41 days) – there was a minor warming in this period when the vortex in the mid-upper stratosphere rapidly decelerated, which may have induced a reflective surface.
- 20-Jan-1996 (47 days) – there was an exceptionally strong and persistent vortex, particularly in the lower-stratosphere, through February.
- 09-Jan-2008 (51 days) – a major SSW occurred in February after an initially very strong vortex at reflection onset.
- 02-Jan-2016 (65 days, longest event in our data) – a very strong vortex had established in the lower stratosphere but there were several upper stratospheric minor warmings.
- 14-Dec-2020 (46 days) – A major SSW occurred on 5 January 2021.

While individually very different, these events were thus all associated with notable stratospheric anomalies, potentially related to the generation of very persistent reflection events. Just as for SSWs, there can be notable differences between individual reflection events, and a nuanced understanding of this variability calls for multiple case studies (including more detailed wave geometry diagnostics as noted above). We believe that this

does not detract from the value of presenting summary statistics, as our own study does, and as indeed has been often done for SSWs.

**Minor comments:**

*Title: I would suggest slightly modifying the title as "Stratospheric downward wave reflection events...... ". The reflection can be either in the vertical or horizontal direction.*

We have changed this as suggested.

*L24-26: Planetary wave patterns are also large-scale meteorological patterns. Please modify this.*

We would argue that the two are partly overlapping, but not synonymous. For example, the Harnik et al. (2016) circumglobal wave pattern would likely not fit the definition of large-scale meteorological pattern given by Loikith et al. (2017): "LSMPs are synoptic-scale patterns defined in terms of key meteorological variables". While we appreciate that other authors may use the terms "planetary wave patterns" and "large-scale meteorological patterns" differently or interchangeably, we therefore see a benefit in introducing these two terminologies separately.

Loikith, P. C., Lintner, B. R., & Sweeney, A. (2017). Characterizing Large-Scale Meteorological Patterns and Associated Temperature and Precipitation Extremes over the Northwestern United States Using Self-Organizing Maps, *Journal of Climate*, *30*(8), 2829-2847.

*L50: It would help the readers to understand what factors controlling the downward wave reflection events. See general comment #4.*

We understand that the Reviewer is wondering whether there may be some large-scale controls modulating the occurrence of the wave reflection events, analogous to the link between the QBO phase and SSWs (McIntyre, 1982). As stated in the reply to general comment #4, we now discuss this point in Sect. 6.

We have additionally tested the link between reflection events and QBO phase, using the monthly mean zonal wind values derived from daily Singapore radiosondes and provided by NASA/GSFC, and defining QBOe and QBOw following Lu *et al.* (2018). Only 11 of our 44 reflection events onset during a QBOe month, while 30 onset during a QBOw month. Similarly, 11 of our 44 reflection events peak during a QBOe month, while 31 peak during a QBOw month. The remaining 3 or 2 events occur during neutral QBO months. There is thus a clear preference for stratospheric reflection events to occur during QBOw, consistent with the strengthened polar vortex observed in association with reflection events (see Figs. A4 and R1 in the replies to Reviewer #1) and the findings from Kretschmer et al. (2018). We

note, however, that while SSWs show clear decadal fluctuations in both occurrence and their link to the QBO (e.g. Lu *et al.*, 2008), the timeseries of reflective event occurrence (Fig. A2b) does not display any evident low–frequency modulation.

McIntyre, M. E. (1982), How well do we understand the dynamics of stratospheric warmings? *J. Meteorol. Soc. Jpn.*, 60, 37–65.

Kretschmer, M., Cohen, J., Matthias, V., Runge, J., & Coumou, D. (2018). The different stratospheric influence on cold-extremes in Eurasia and North America. *npj Climate and Atmospheric Science*, *1*(1), 1-10.

Lu, H., M. P. Baldwin, L. J. Gray, and M. J. Jarvis, 2008: Decadal-scale changes in the effect of the QBO on the northern stratospheric polar vortex. *J. Geophys. Res.*, **113**, D10114, doi:10.1029/2007JD009647.

*L52: "One potential reason is the difficulty in diagnosing reflection events (see the discussion in Matthias and Kretschmer, 2020)." I don't think this is the main reason. I think the main reason is the underlying dynamics that are still unclear, especially from the perspective of eddy-mean flow interaction (e.g., baroclinic eddy feedback in the presence of the downward reflection) and energy transfer (e.g., Harnik 2009, Lubis et al., 2017).*

We agree that the lack of a full understanding of the dynamical processes underlying wave reflection complicates the analysis of wave reflection events (see also our answer to comment #1). We have added this perspective to our introduction, including the references suggested by the Reviewer.

*L55-59: "Matthias and Kretschmer (2020) introduced a simple index to identify wave reflection events based on anomalous lower-stratospheric poleward eddy-heat flux over Siberia and Canada". Matthias and Kretschmer (2020) in fact used the same measure of downward WAF as originally used by Kodera's back in 2008 (see Figure 2a in Kodera et al., (2008), with the same level (i.e., 100 hPa) and the same regional locations associated with the upward and downward WAF). It would be nice to acknowledge the work of Kodera et al., (2008) too here.*
*Kodera, K., Mukougawa, H., and Itoh, S. (2008), Tropospheric impact of reflected planetary waves from the stratosphere, Geophys. Res. Lett., 35, L16806, doi:10.1029/2008GL034575.*

We agree that it is important to acknowledge the work of Kodera and colleagues in this passage, which we now do. Nevertheless, we believe that there are relevant differences between the work by Matthias and Kretschmer (2020) and that by Kodera *et al.* (2008). Indeed, Kodera *et al.* do not use the simple Siberian and Canadian box approach for identifying reflection events, which was the main innovation in the study by Matthias and Kretschmer (2020).

*L74: Why October and November are excluded from the analysis? It would be great to argue this from dynamical perspective of downward reflection events (see the seasonality of DWC in Shaw et al., 2010, Lubis et al., 2017 etc).*

One of the aims of this study is to link stratospheric processes, and specifically wave reflection, to tropospheric circulation and surface impacts. One approach would naturally be to conduct the analysis over the full year. However, this presents obvious difficulties due to both a clear seasonality in stratospheric processes (*e.g.* all satellite-era SSWs have occurred in DJFM, and vortex variability peaks during mid-winter) and a clear seasonality in tropospheric circulation regimes and surface impacts. We have ultimately settled on the choice of DJFM because this is a season conventionally used to define tropospheric weather regimes (see e.g. Casola and Wallace, 2007; Ouzeau *et al.*, 2011; Lucas-Picher *et al.*, 2016; Lee *et al.*, 2019), when the stratosphere-troposphere coupling and vortex variability is largest, and surface cold extremes are most intense (e.g. Table 2c in Walsh *et al.*, 2001). This does not mean that ours is the only possible choice, but we believe it is well-motivated from both a climate dynamics perspective and ease of comparison with previous literature. In the revised manuscript we now mention the possibility of the occurrence of wave reflection events in October and November not covered by our study in Sect. 2.

Casola, J. H., & Wallace, J. M. (2007). Identifying weather regimes in the wintertime 500-hPa geopotential height field for the Pacific–North American sector using a limited-contour clustering technique. *Journal of applied meteorology and climatology*, *46*(10), 1619-1630.

Lucas-Picher, P., Cattiaux, J., Bougie, A., & Laprise, R. (2016). How does large-scale nudging in a regional climate model contribute to improving the simulation of weather regimes and seasonal extremes over North America?. *Climate Dynamics*, *46*(3), 929-948.

Ouzeau, G., Cattiaux, J., Douville, H., Ribes, A., & Saint-Martin, D. (2011). European cold winter 2009–2010: How unusual in the instrumental record and how reproducible in the ARPEGE-Climat model?. *Geophysical Research Letters*, *38*(11).

Walsh, J. E., Phillips, A. S., Portis, D. H., & Chapman, W. L. (2001). Extreme cold outbreaks in the United States and Europe, 1948–99. *Journal of climate*, *14*(12), 2642-2658.

*L102: I would specifically mention that your work only focus on the impacts over the North America sector.*

We have added this to the text as suggested.

*L111: Can you check the sensitivity with respect to the size of the regional boxes used to define the index.*

We checked the RI for different sizes of regional boxes. We tested enlarging the two boxes by ten degrees longitude in either direction in turn, and by 10 degrees in both longitudinal directions for both the Siberian and Canadian boxes. This leads to five additional RIs, which differ from the original RI by 0.11 on average and 0.9 as maximum over all available years.

All RIs are further highly correlated with the original RI (corr >0.99). There are times when the RI indices are sufficiently different as to affect the detection of reflection events, yet these are rare. Depending on the chosen shift in domain, there are between 0 and 2 sequences of more than 10 reflective days that are detected using the boxes of the paper but not when using the modified boxes. The boxes used in the paper in fact tend to detect fewer prolonged reflection periods than some of the alternative boxes tested here, and in this sense may be seen as a conservative choice. To illustrate this, we present in Fig. R11 the results for both a change in the Siberian domain (extension by 10 degrees to the west) with no change in the Canadian domain and for a change in the Canadian domain (extension by 10 degrees to the east) with no change in the Siberian domain. The blue pixels represent days which are reflective using the boxes of the manuscript but not when using the extended boxes. The red pixels represent days which are reflective in the extended boxes but not when using the boxes defined in the manuscript. In the first case, there are two prolonged periods when the RI as defined in the paper does detect reflection while the RI with the modified domain does not. In the second case, there is one prolonged period when the RI as defined in the paper does detect reflection while the RI with the modified domain does not, and two periods when the converse happens.

As has been illustrated in the various sensitivity tests on surface temperature anomalies conducted in Figs. R2–R5, R9 and R12–R15, even relatively large changes in our sample size do not alter the qualitative footprint of the surface temperature anomalies. Given this and the rarity of temporally persistent discrepancies in the detection of reflection events, we conclude that moderate changes on the regional boxes used to define reflection events do not affect our qualitative conclusions.

[Figure]

*Fig. R11: Reflection days calculated using: (a) Sib: 130°-200°E and Can: 230°-280°E; and (b) Sib: 140°-200°E and Can: 230°-290°E. The blue pixels represent days which are reflective using the boxes of the manuscript but are classified as non-reflective when using the extended boxes. The red pixels represent days which are reflective using the extended boxes but are classified as non-reflective when using the boxes defined in the manuscript. The y-axis shows years from 1980 to 2021 and the x-axis shows days of the year, from 1st December of the year before the one shown on the y-axis to 30th March of the year shown on the y-axis.*

*L128: " ..while negative values indicate downward propagation." Add the following sentence at the end of this "(assuming the wave-activity density is positive definite)".*

We have added this to the text as suggested.

*L134: It may be not true for the whole domain over North America since you are using the anomalous eddy heat flux. See my general comment #5.*

As noted in our response to comment #5, Fig. 1d does show the *full* v'T' field, and thus we believe our interpretation is correct.

*L147: "Finally, following Matthias and Kretschmer (2020) we apply a persistence criterion of 10 days". Would be the results sensitive with the choice of this threshold? Please clarify.*

The persistence criterion was introduced to filter out the events showing a notable tropospheric impact, which form the focus of the present study. While any specific choice of persistence in days is to some degree arbitrary, 10 days was selected as a good balance between long persistence and sample size. Also, this is the typical time-scale used to refer to "beyond" weather events (for example, ECMWF's high-resolution deterministic forecast is run up to a 10-day lead time), making it a natural cut-off. It is further supported by the fact that the dynamics of wintertime surface temperature extremes typically develop on weekly or longer timescales (e.g. Messori *et al.*, 2016).

Changing the persistence threshold naturally affects the number of reflective events we detect. We have tested thresholds of 7 and 14 days, for which we obtain 60 and 30 events respectively. Nonetheless, the qualitative results concerning the surface anomalies associated with the reflection events are consistent with those obtained for the 10 day threshold used in the study. While there is some modulation of the magnitude of the temperature anomalies, with the more persistent reflection events locally leading to stronger negative anomalies, the difference is generally small. This points to the conclusions we draw from our analysis not being overly sensitive to the chosen 10 days persistence threshold. We show in Figs. R12–R15 below the results corresponding to Figs. 4c, d and Fig. 5 in the main text. We now mention this in the revised text in Sect. 3.

[Figure]

*Fig. R12. Same as Fig. 4c, d in the main paper, but for a 7-day persistence threshold (60 Events).*

[Figure]

*Fig. R13. Same as Fig. 4c, d in the main paper, but for a 14-day persistence threshold (30 Events).*

[Figure]

*Fig. R14. Same as Fig. 5 in the main paper, but for a 7-day persistence threshold.*

[Figure]

*Fig. R15. Same as Fig. 5 in the main paper, but for a 14-day persistence threshold.*

*L164: "we analyze the tropospheric evolution associated with the 44 reflection events". How many events are related to SSW event? Please see my general comment #3.*

We have now updated Table A1. 9 out of 44 reflection events correspond to a SSW, but only three are followed by an SSW within 15 days of the reflection event onset – and none within the first nine days when the primary response to the reflection event occurs. We now discuss

this in the main text. We refer the Reviewer to our reply to their major comment #3 for a more detailed discussion on this point.

*Figure 6. Can you explain dynamically how does the wave reflection event cause a stretching tropospheric jet, hence a severe cold spell over North America? Can you quantify this? I think the elongation of tropospheric jet (i.e., wavier) is of importance to the cold spell over North America, which may provide useful insight to the forecast of cold spells over North America.*

In Fig. 6 we do not explicitly argue for a stretching of the tropospheric jet. Indeed, defining the location of the tropospheric jet is a whole field of research in of itself (ranging from simpler zonal-mean indices such as that of Woollings *et al.* (2010) to still simple yet latitudinally varying indices (e.g. Faranda et al., 2019), to complex "jet core" indices (e.g. Spensberger et al., 2017). Quantifying jet sinuosity is an equally wide field (e.g. Martin et al., 2016; Cattiaux et al., 2016), and one which has elicited considerable controversy in the community, especially in the context of mid-high latitude linkages. Systematically quantifying the role of jet location and sinuosity on North American cold spells is thus well beyond the current scope.

We further note that in forecasting applications, weather regimes are usually preferred to addressing the jet's location and sinuosity (e.g. Lee et al., 2022 and references therein, but also the work of Ferranti *et al.*, 2015 and Matsueda and Palmer, 2018 on flow-dependent predictability). What has been shown is that planetary wave patterns, associated with meanders of the jet, can provide useful statistical predictability for North American cold spells (Harnik *et al.*, 2016). From the point of view of how wave reflection can influence tropospheric circulation patterns, some hints may be gleaned from the vertical wave structure in Fig. 12, showing an eastward-tilting, downward-propagating wave structure over the Canada node. We agree that this is an important point, and have added a summary of the above discussion to Sect. 6 in the main text.

Woollings, T., Hannachi, A., & Hoskins, B. (2010). Variability of the North Atlantic eddy-driven jet stream. *Quarterly Journal of the Royal Meteorological Society*, *136*(649), 856-868.

Frame, T. H. A., Methven, J., Gray, S. L., & Ambaum, M. H. P. (2013). Flow-dependent predictability of the North Atlantic jet. *Geophysical Research Letters*, *40*(10), 2411-2416.

Ferranti, L., Corti, S., & Janousek, M. (2015). Flow-dependent verification of the ECMWF ensemble over the Euro-Atlantic sector. *Quarterly Journal of the Royal Meteorological Society*, *141*(688), 916-924.

Martin, J. E., Vavrus, S. J., Wang, F., & Francis, J. A. (2016). Sinuosity as a measure of middle tropospheric waviness 6. *Journal of Climate*, *25*, 26.

Harnik, N., Messori, G., Caballero, R., & Feldstein, S. B. (2016). The circumglobal North American wave pattern and its relation to cold events in eastern North America. *Geophysical Research Letters*, *43*(20), 11-015.

Cattiaux, J., Peings, Y., Saint‑Martin, D., Trou‑Kechout, N., & Vavrus, S. J. (2016). Sinuosity of midlatitude atmospheric flow in a warming world. *Geophysical Research Letters*, *43*(15), 8259-8268.

Matsueda, M., & Palmer, T. N. (2018). Estimates of flow‑dependent predictability of wintertime Euro‑Atlantic weather regimes in medium‑range forecasts. *Quarterly Journal of the Royal Meteorological Society*, *144*(713), 1012-1027.

Faranda, D., Sato, Y., Messori, G., Moloney, N. R., & Yiou, P. (2019). Minimal dynamical systems model of the northern hemisphere jet stream via embedding of climate data. *Earth System Dynamics*, *10*(3), 555-567.

Spensberger, C., Spengler, T., & Li, C. (2017). Upper-tropospheric jet axis detection and application to the boreal winter 2013/14. *Monthly Weather Review*, *145*(6), 2363-2374.

*L260-L265: I am afraid that your signals are mixed with the SSW events. Please do check. See my general comment #3.*

As shown in the updated table A1, only a small number of our reflection events are associated with SSWs, none of which occur during the primary response period within the first week to 10 days of the reflection onset. Fig R9 in our reply to major comment #3 also suggests that the surface temperature footprint we obtain for our reflection events is not due to concomitant SSWs.

*L321-L323: Do you have any cases where a downward reflection event is not preceded by the PT regime, but still have a strong influence on surface temperature? Or a case when you have reflection but there is no significant impact on the surface temperature. I am wondering whether the PT regime is a robust tropospheric precursor of the downward reflection event.*

We have combined the information on the tropospheric precursors of the reflection events with the information on the surface temperature anomalies following the reflection events. Of our 44 events, 33 show a PT regime on at least one day in the three days prior to the event onset. Of these 33 events, 24 are cold, 7 are warm and 2 are neutral. Of the remaining 11 events, 4 are cold, 5 are warm and 2 are neutral. Thus, (albeit with a caution on the limited sample size) events not preceded by the PT regime, which we consider in our analysis as the "canonical" regime associated with the onset of the reflection events, indeed show a weaker association to cold surface temperatures as the event develops relative to reflection events associated with the PT regime. We believe that this, coupled with the geographical composites we show in the paper, supports the robustness of the PT regime as a tropospheric precursor to reflection events associated with cold spells over North America.

*L325: Very good point but without exploring this, it remains elusive if wave reflection events are the main cause of the cold spells.*

In reply to the Reviewer's other comments, we have performed additional analyses to better understand which role the reflection events play in favouring surface cold spells. These have resulted in the addition of numerous discussion points and a clearer acknowledgment of the role of tropospheric precursors to reflection events and of other stratospheric processes, such as SSWs, which may confound the surface signal. The additions on these points are mainly concentrated in Sect. 6 of the revised manuscript, and are covered in our replies to several of the above comments. However, we would argue that even without these additions our analysis does show a robust association between reflection events and surface cold spells, as stated on l. 326, since robust co-occurrence can be present regardless of the direction of causality. Again, we argue that such a statistical association can in itself be valuable from a predictability perspective.

---

## Referee Report (RR1)

Review of manuscript WCD-2022-18: "**Stratospheric Downward Wave Reflection Events Modulate North American Weather Regimes and Cold Spells**", by Messori et al.

The authors addressed mostly my main concerns by conducting a further analysis on the eddy height field and the North American weather regime transitions (including the new figures 9 and 12). Although the authors did not discuss the direct relationship between the reflections and the degree of coldness from the dynamical perspective (i.e., eddy-mean flow interaction and enstrophy/energy transfer between wave and mean flow during reflection events), this work is still valuable and to some extent gives important insight into the role of wave reflection events in sub-seasonal wintertime forecasts via a top-down mechanism. I really enjoyed reading this paper, for me this is one of the most comprehensive papers showing the robust statistical link between the wave reflection events and North American weather regimes/cold spells. As such, I recommend that the paper be accepted with a minor revision:

**Minor comments:**

- Fig. 12, I think it's better to superimpose the eddy heights with the 2D Plumb's fluxes to better show the downward and upward energy propagation during these reflection events (the energy flux is equal to the group velocity times wave-activity density).
- Table A1, can you clarify how many reflection events that occurred after the SSW events? The reason I'm asking this because in some cases the reflections occur after SSW events, not overlapping (i.e., reflective SSW events).

Best wishes,
Sandro W. Lubis

---

## Author Response (AR2)

**Reviewer #1**

The authors addressed mostly my main concerns by conducting a further analysis on the eddy height field and the North American weather regime transitions (including the new figures 9 and 12). Although the authors did not discuss the direct relationship between the reflections and the degree of coldness from the dynamical perspective (i.e., eddy-mean flow interaction and enstrophy/energy transfer between wave and mean flow during reflection events), this work is still valuable and to some extent gives important insight into the role of wave reflection events in sub-seasonal wintertime forecasts via a top-down mechanism. I really enjoyed reading this paper, for me this is one of the most comprehensive papers showing the robust statistical link between the wave reflection events and North American weather regimes/cold spells. As such, I recommend that the paper be accepted with a minor revision.

We would like to thank Sandro Lubis for his constructive feedback throughout the review process and his positive outlook on the latest version of our manuscript. We copy below the minor comments in *italics* and our replies in blue.

**Minor comments**
*1. Fig. 12, I think it's better to superimpose the eddy heights with the 2D Plumb's fluxes to better show the downward and upward energy propagation during these reflection events (the energy flux is equal to the group velocity times wave-activity density).*

We have updated the figure as suggested, and now show the vertical and zonal components of the Plumb wave activity flux.

*2. Table A1, can you clarify how many reflection events that occurred after the SSW events? The reason I'm asking this because in some cases the reflections occur after SSW events, not overlapping (i.e., reflective SSW events).*

Only three of the 35 reflection events that do not overlap with a SSW are preceded within 20 days by a SSW (events no. 3, 24 and 25 in Table A1). We picked 20 days as it is the standard SSW event separation criterion (*e.g.*, Charlton and Polvani, 2007). We now mention this in Sect. 6. We believe this does not alter the conclusions presented in the study.

Charlton, A. J., & Polvani, L. M. (2007). A New Look at Stratospheric Sudden Warmings. Part I: Climatology and Modeling Benchmarks, *Journal of Climate*, *20*(3), 449-469. Retrieved Aug 18, 2022, from https://journals.ametsoc.org/view/journals/clim/20/3/jcli3996.1.xml

**Reviewer #2**

The authors did a great job in the revision. I'm satisfied with most of the revisions. Now I only have a few minor comments before the final acceptance.

We would like to thank Pengfei Zhang for his constructive feedback throughout the review process and his positive outlook on the latest version of our manuscript. We copy below the minor comments in *italics* and our replies in blue.

*1. For the discussion of Fig. A4, the authors argue that:*
*In Sect. 3: "The onset of these reflection events is associated with an anomalously strong stratospheric polar vortex (negative Z10 anomalies over the central Arctic, Fig. A4), consistent with the expected favorable conditions for wave reflection."*

*According to the fig. A4, by comparing the change from d-3 to d0 and the following temporal evolution, it seems the polar cap averaged polar vortex is weakening, and the vortex is shifting. The polar cap averaged vortex may be a negative anomaly on d0, but it has already started the weakening variation (see d-3 vs d0). Thus, according to the figures, I think it would be more appropriate to state that the reflection event is associated with a slightly(?) stronger but a weakening of the stratospheric polar vortex. The authors may confirm the time series of zonal mean zonal wind at 60N at 10hPa or polar cap Z at 10hPa (north of 60 or 65N) anomalies to make the final conclusion. Both are conventional indices to measure the overall status of the polar vortex.*

[Figure]

*Figure R1: Evolution of the 10 hPa 60°N zonal-mean zonal wind (a) absolute values and (b) anomalies for the 44 reflection events. The thick red line denotes the average over all events. Red shading indicates the 95\% confidence interval on the mean assessed as described in Sect. 2.*

Figure R1 (added to the paper as the new Fig. A5) shows the evolution of the 10 hPa 60°N zonal-mean zonal winds during reflection events. While the polar vortex does show a weakening at positive lags, it is roughly constant in intensity and stronger than climatology between day -3 and day 0. We thus believe that our statement that the onset of the reflection events is associated with an anomalously strong stratospheric polar vortex holds. We now additionally mention the weakening at positive lags and return to climatological levels in both Sect. 3 and Sect. 6. We further briefly discuss in Sect. 6 that, beyond day +3, there is essentially no signal in zonal wind strength on average during the reflection events. This suggests that reflection events do not induce a specific U10 change as they develop.

*2. The statement in Sect. 6: "Significant negative 10 hPa geopotential height anomalies persist over the polar region for over 10 days following the onset of the reflection events (Fig. A4)."*

*Similarly, I don't think this is a correct argument. If we calculated the polar region averaged Z anomalies at 10hPa, I would guess the values during these 10 days could be negative first followed by near zero or even positive (a transition). The negative anomalies within the vortex actually denote the shifting of the vortex. Shifting is a kind of weakening from the view of the area average over the polar cap or the zonal mean zonal wind at 60N (the overall status of polar vortex).*

As shown in Fig. R1 above, the polar vortex is on average stronger than usual during the onset of the reflection events and up to roughly day +3. The vortex then returns to climatological values and the 10 hPa 60°N zonal-mean zonal wind anomalies remain close to 0 up to day +24. We have now rephrased the text to clarify that there is a vortex shifting and weakening (as measured by the zonal-mean zonal wind). We additionally contextualise this relative to the wave-1 stratospheric pattern discussed in Ding et al. (2022). However, the key point we wanted to make is that the negative Z10 anomalies, albeit shifted geographically, persist even at positive lags and do not resemble the picture typically associated with major SSWs.

Ding, X., Chen, G., Sun, L., & Zhang, P. (2022). Distinct North American cooling signatures following the zonally symmetric and asymmetric modes of winter stratospheric variability. *Geophysical Research Letters*, 49, e2021GL096076. https://doi.org/10.1029/2021GL096076

*3. After the last round of review, I noticed a paper could be helpful for some discussion in the current study. See Guan et al. 2020 (https://doi.org/10.1175/JCLI-D-20-0096.1). (BTW, I'm not the co-author.) Guan et al. (2020) reported a weather regime transition in North America at the subseasonal timescale, which also involves wave reflection.*

Thank you for having suggested this relevant paper. We now cite it in the introduction and discuss it in greater detail in Sect. 6.

---

## Author Response (AR3)

Dear Daniela, we have edited the manuscript according to your suggestions, and have included a link to a repository with doi where the weather regimes data may be accessed publicly.

Best,

Gabriele also on behalf of all co-authors